# Multinomial Logistic Regression: Asymptotic Normality on Null Covariates in High-Dimensions

**Kai Tan**
Department of Statistics
Rutgers University
Piscataway, NJ 08854
`kai.tan@rutgers.edu`

**Pierre C. Bellec**
Department of Statistics
Rutgers University
Piscataway, NJ 08854
`pierre.bellec@rutgers.edu`

## Abstract

This paper investigates the asymptotic distribution of the maximum-likelihood estimate (MLE) in multinomial logistic models in the high-dimensional regime where dimension and sample size are of the same order. While classical large-sample theory provides asymptotic normality of the MLE under certain conditions, such classical results are expected to fail in high-dimensions as documented for the binary logistic case in the seminal work of Sur and Candès [2019]. We address this issue in classification problems with 3 or more classes, by developing asymptotic normality and asymptotic chi-square results for the multinomial logistic MLE (also known as cross-entropy minimizer) on null covariates. Our theory leads to a new methodology to test the significance of a given feature. Extensive simulation studies on synthetic data corroborate these asymptotic results and confirm the validity of proposed p-values for testing the significance of a given feature.

## 1 Introduction

Multinomial logistic modeling has become a cornerstone of classification problems in machine learning, as witnessed by the omnipresence of both the cross-entropy loss (multinomial logistic loss) and the softmax function (gradient of the multinomial logistic loss) in both applied and theoretical machine learning. We refer to Cramer [2002] for an account of the history and early developments of logistic modeling.

Throughout, we consider a classification problem with $K + 1$ possible labels where $K$ is a fixed constant. This paper tackles asymptotic distributions of multinomial logistic estimates (or cross-entropy minimizers) in generalized linear models with moderately high-dimensions, where sample size $n$ and dimension $p$ have the same order, for instance $n, p \to +\infty$ simultaneously while the ratio $p/n$ converges to a finite constant. Throughout the paper, let $[n] = \{1, 2, \ldots, n\}$ for all $n \in \mathbb{N}$, and $I\{\text{statement}\}$ be the 0-1 valued indicator function, equal to 1 if statement is true and 0 otherwise (e.g., $I\{y_i = 1\}$ in the next paragraph equals 1 if $y_i = 1$ holds and 0 otherwise).

**The case of binary logistic regression.** Let $\rho(t) = \log(1 + e^t)$ be the logistic loss and $\rho'(t) = 1/(1 + e^{-t})$ be its derivative, often referred to as the sigmoid function. In the current moderately-high dimensional regime where $n, p \to +\infty$ with $p/n \to \kappa > 0$ for some constant $\kappa$, recent works [Candès and Sur, 2020, Sur and Candès, 2019, Zhao et al., 2022] provide a detailed theoretical understanding of the behavior of the logistic Maximum Likelihood Estimate (MLE) in binary logistic regression models. Observing independent observations $(x_i, y_i)_{i \in [n]}$ from a logistic model defined as $\mathbb{P}(y_i = 1 | x_i) = \rho'(x_i^T \beta)$ where $x_i \sim N(\mathbf{0}, n^{-1} I_p)$, and $\lim_{n \to \infty} \|\beta\|^2 / n = \gamma^2$ for a constant $\gamma$ for the limiting squared norm of the unknown regression vector $\beta$. These works prove that the behavior of the MLE $\hat{\beta} = \arg\min_{b \in \mathbb{R}^p} \sum_{i=1}^n \rho(x_i^T b) - I\{y_i = 1\} x_i^T b$ is summarized by the solution

$(\alpha_*, \sigma*, \lambda_*)$ of the system of three equations

$$
\begin{cases}
\sigma^2 & = \frac{1}{\kappa^2}\mathbb{E}[2\rho'(\gamma Z_1)(\lambda\rho'(\mathrm{prox}_{\lambda\rho}(-\alpha\gamma Z_1 + \sqrt{\kappa}\sigma Z_2)))^2] \\
0 & = \mathbb{E}[\rho'(\gamma Z_1)\lambda\rho'(\mathrm{prox}_{\lambda\rho}(-\alpha\gamma Z_1 + \sqrt{\kappa}\sigma Z_2))] \\
1-\kappa & = \mathbb{E}[2\rho'(\gamma Z_1)/(1 + \lambda\rho''(\mathrm{prox}_{\lambda\rho}(-\alpha\gamma Z_1 + \sqrt{\kappa}\sigma Z_2)))]
\end{cases}, \tag{1.1}
$$

where $(Z_1, Z_2)$ are i.i.d. $N(0,1)$ random variables and the proximal operator is defined as $\mathrm{prox}_{\lambda\rho}(z) = \arg\min_{t\in\mathbb{R}}\{\lambda\rho(t) + (t-z)^2/2\}$. The system (1.1) characterize, among others, the following behavior of the MLE $\hat{\beta}$: for almost any $(\gamma, \kappa)$, the system admits a solution if and only if $\hat{\beta}$ exists with probability approaching one and in this case, $\|\hat{\beta}\|^2/n$ and $\|\hat{\beta} - \beta\|^2/n$ both have finite limits that may be expressed as simple functions of $(\alpha_*, \sigma_*, \lambda_*)$, and for any feature $j \in [p]$ such that $\beta_j = 0$ (i.e., $j$ is a null covariate), the $j$-th coordinate of the MLE satisfies

$$
\hat{\beta}_j \xrightarrow{\mathrm{d}} N(0, \sigma_*^2).
$$

The proofs in Sur and Candès [2019] are based on approximate message passing (AMP) techniques; we refer to Berthier et al. [2020], Feng et al. [2022], Gerbelot and Berthier [2021] and the references therein for recent surveys and general results. More recently, Zhao et al. [2022] extended the result of Sur and Candès [2019] from isotropic design to Gaussian covariates with an arbitrary covariance structure: if now $x_i \sim N(\mathbf{0}, \Sigma)$ for some positive definite $\Sigma$ and $\lim_{n,p\to+\infty} \beta^T\Sigma\beta = \kappa$, null covariates $j \in [p]$ (in the sense that $y_i$ is independent of $x_{ij}$ given $(x_{ik})_{k\in[p]\setminus\{j\}}$) of the MLE satisfy

$$
(n/\Omega_{jj})^{1/2}\hat{\beta}_j \xrightarrow{\mathrm{d}} N(0, \sigma_*^2), \tag{1.2}
$$

where $\sigma_*$ is the same solution of (1.1) and $\Omega = \Sigma^{-1}$. Zhao et al. [2022] also obtained asymptotic normality results for non-null covariates, that is, features $j \in [p]$ such that $\beta_j \neq 0$. The previous displays can be used to test the null hypothesis $H_0 : y_i$ is independent of $x_{ij}$ given $(x_{ik})_{k\in[p]\setminus\{j\}}$ and develop the corresponding p-values if $\sigma_*$ is known; in this binary logistic regression model the ProbeFrontier [Sur and Candès, 2019] and SLOE Yadlowsky et al. [2021] give means to estimate the solutions $(\alpha_*, \sigma_*, \lambda_*)$ of system (1.1) without the knowledge of $\gamma$. Mai et al. [2019] studied the performance of Ridge regularized binary logistic regression in mixture models. Salehi et al. [2019] extended Sur and Candès [2019] to separable penalty functions. Bellec [2022] derived asymptotic normality results similar to (1.2) in single-index models including binary logistic regression without resorting to the system (1.1), showing that for a null covariate $j \in [p]$ in the unregularized case that

$$
(n/\Omega_{jj})^{1/2}(\hat{v}/\hat{r})\hat{\beta}_j \xrightarrow{\mathrm{d}} N(0, 1) \tag{1.3}
$$

where $\hat{v} = \frac{1}{n}\sum_{i=1}^n \rho''(x_i^T\hat{\beta}) - \rho''(x_i^T\hat{\beta})^2 x_i^T[\sum_{l=1}^n x_l\rho''(x_l^T\hat{\beta})x_l^T]^{-1}x_i$ is scalar and so is $\hat{r}^2 = \frac{1}{n}\sum_{i=1}^n(I\{y_i = 1\} - \rho'(x_i^T\hat{\beta}))^2$. In summary, in this high dimensional binary logistic model,

  (i) The phase transition from Candès and Sur [2020] splits the $(\gamma, \kappa)$ plane into two connected components: in one component the MLE does not exist with high probability, in the other component the MLE exists and $\|\Sigma^{1/2}\hat{\beta}\|^2$ is bounded with high probability (boundedness is a consequence of the fact that $\|\Sigma^{1/2}\hat{\beta}\|^2$ or $\|\Sigma^{1/2}(\hat{\beta} - \beta)\|^2$ admit finite limits);

  (ii) In the component of the $(\gamma, \kappa)$ plane where the MLE exists, for any null covariate $j \in [p]$, the asymptotic normality results (1.2)-(1.3) holds.

**Multiclass classification.** The goal of this paper is to develop a theory for the asymptotic normality of the multinomial logistic regression MLE (or cross-entropy minimizer) on null covariates when the number of classes, $K+1$, is greater than 2 and $n, p$ are of the same order. In other words, we aim to generalize results such as (1.2) or (1.3) for three or more classes. Classification datasets with 3 or more classes are ubiquitous in machine learning (MNIST, CIFAR to name a few), which calls for such multiclass generalizations. In Gaussian mixtures and logistic models, Thrampoulidis et al. [2020] derived characterizations of the performance of of least-squares and class-averaging estimators, excluding cross-entropy minimizers or minimizers of non-linear losses. Loureiro et al. [2021] extended Sur and Candès [2019], Zhao et al. [2022], Salehi et al. [2019] to multiclass classification problems in a Gaussian mixture model, and obtained the fixed-point equations that characterize the performance and empirical distribution of the minimizer of the cross-entropy loss plus a convex regularizer. In the same vein as Loureiro et al. [2021], Cornacchia et al. [2022] studied the limiting

fixed-point equations in a multiclass teacher-student learning model where labels are generated by a noiseless channel with response $\arg\min_{k\in\{1,\dots,K\}} x_i^T \beta_k$ where $\beta_k \in \mathbb{R}^p$ is unknown for each class $k$. These two aforementioned works assume a multiclass Gaussian mixture model, which is different than the normality assumption for $x_i$ used in the present paper. More importantly, these results cannot be readily used for the purpose testing significant covariates (cf. (1.10) below) since solving the fixed-point equations require the knowledge of several unknown parameters, including the limiting spectrum of the mixture covariances and empirical distributions of the mixture means (cf. for instance Corollary 3 in Loureiro et al. [2021]). In the following sections, we fill this gap with a new methodology to test the significance of covariates. This is made possible by developing new asymptotic normality results for cross-entropy minimizers that generalize (1.3), without relying on the low-dimensional fixed-point equations.

**Notation.**    Throughout, $I_p \in \mathbb{R}^{p\times p}$ is the identity matrix, for a matrix $A \in \mathbb{R}^{m\times n}$, $A^T$ denotes the transpose of $A$, $A^\dagger$ denotes the Moore-Penrose inverse of $A$. If $A$ is psd, $A^{1/2}$ denotes the unique symmetric square root, i.e., the unique positive semi-definite matrix such that $(A^{1/2})^2 = A$. The symbol $\otimes$ denotes the Kronecker product of matrices. Given two matrices $A \in \mathbb{R}^{n\times k}, B \in \mathbb{R}^{n\times q}$ with the same number or rows, $(A, B) \in \mathbb{R}^{n\times(k+q)}$ is the matrix obtained by stacking the columns of $A$ and $B$ horizontally. If $v \in \mathbb{R}^n$ is a column vector with dimension equal to the number of rows in $A$, we construct $(A, v) \in \mathbb{R}^{n\times(k+1)}$ similarly. We use $\mathbf{0}_n$ and $\mathbf{1}_n$ to denote the all-zeros vector and all-ones vector in $\mathbb{R}^n$, respectively; we do not bold vectors and matrices other than $\mathbf{0}_n$ and $\mathbf{1}_n$. We may omit the subscript giving the dimension if clear from context; e.g., in $I_{K+1} - \frac{\mathbf{1}\mathbf{1}^T}{K+1}$ the vector $\mathbf{1}$ is in $\mathbb{R}^{K+1}$. The Kronecker product between two matrices is denoted by $\otimes$ and $\mathrm{vec}(M) \in \mathbb{R}^{nd}$ is the vectorization operator applied to a matrix $M \in \mathbb{R}^{n\times d}$. For an integer $K \geq 2$ and $\alpha \in (0,1)$, the quantile $\chi_K^2(\alpha)$ is the unique real number satisfying $\mathbb{P}(W > \chi_K^2(\alpha)) = \alpha$ where $W$ has a chi-square distribution with $K$ degrees of freedom. The symbols $\xrightarrow{d}$ and $\xrightarrow{\mathbb{P}}$ denote convergence in distribution and in probability.

Throughout, *classical asymptotic regime* refers to the scenario where the feature dimension $p$ is fixed and the sample size $n$ goes to infinity. In contrast, the term *high-dimensional regime* refers to the situation where $n$ and $p$ both tend to infinity with the ratio $p/n$ converging to a limit smaller than 1.

## 1.1  Multinomial logistic regression

Consider a multinomial logistic regression model with $K+1$ classes. We have $n$ i.i.d. data samples $\{(x_i, \mathsf{y}_i)\}_{i=1}^n$, where $x_i \in \mathbb{R}^p$ is the feature vector and $\mathsf{y}_i = (\mathsf{y}_{i1}, \dots, \mathsf{y}_{i(K+1)})^T \in \mathbb{R}^{K+1}$ is the response. Each response $\mathsf{y}_i$ is the one-hot encoding of a single label, i.e., $\mathsf{y}_i \in \{0,1\}^{K+1}$ with $\sum_{k=1}^{K+1} \mathsf{y}_{ik} = 1$ such that $\mathsf{y}_{ik} = 1$ if and only if the label for $i$-th observation is $k$. A commonly used generative model for $\mathsf{y}_i$ is the multinomial regression model, namely

$$\mathbb{P}(\mathsf{y}_{ik} = 1 | x_i) = \frac{\exp(x_i^T \mathsf{B}^* e_k)}{\sum_{k'=1}^{K+1} \exp(x_i^T \mathsf{B}^* e_{k'})}, \quad k \in \{1, 2, \dots, K+1\} \tag{1.4}$$

where $\mathsf{B}^* \in \mathbb{R}^{p\times(K+1)}$ is an unknown logistic model parameter and $e_k \in \mathbb{R}^{K+1}, e_{k'} \in \mathbb{R}^{K+1}$ are the $k$-th and $k'$-th canonical basis vectors. The MLE for $\mathsf{B}^*$ in the model (1.4) is any solution that minimizes the cross-entropy loss,

$$\hat{\mathsf{B}} \in \arg\min_{\mathsf{B}\in\mathbb{R}^{p\times(K+1)}} \sum_{i=1}^n \mathsf{L}_i(\mathsf{B}^T x_i), \tag{1.5}$$

where $\mathsf{L}_i : \mathbb{R}^{K+1} \to \mathbb{R}$ is defined as $\mathsf{L}_i(\mathsf{u}) = -\sum_{k=1}^{K+1} \mathsf{y}_{ik}\mathsf{u}_k + \log\sum_{k'=1}^{K+1}\exp(\mathsf{u}_{k'})$. If the solution set in (1.5) is non-empty, we define for each observation $i \in [n]$ the vector of predicted probabilities $\hat{\mathsf{p}}_i = (\hat{\mathsf{p}}_{i1}, \dots, \hat{\mathsf{p}}_{i(K+1)})^T$ with

$$\hat{\mathsf{p}}_{ik} \overset{\text{def}}{=} \mathbb{P}(\hat{\mathsf{y}}_{ik} = 1) = \frac{\exp(x_i^T \hat{\mathsf{B}} e_k)}{\sum_{k'=1}^{K+1} \exp(x_i^T \hat{\mathsf{B}} e_{k'})} \qquad \text{for each } k \in \{1, \dots, K+1\}. \tag{1.6}$$

Our results will utilize the gradient and Hessian of $\mathsf{L}_i$ evaluated at $\hat{\mathsf{B}}^T x_i$, denoted by

$$\mathsf{g}_i \overset{\text{def}}{=} \nabla\mathsf{L}_i(\hat{\mathsf{B}}^T x_i) = -\mathsf{y}_i + \hat{\mathsf{p}}_i, \qquad \mathsf{H}_i \overset{\text{def}}{=} \nabla^2\mathsf{L}_i(\hat{\mathsf{B}}^T x_i) = \mathrm{diag}(\hat{\mathsf{p}}_i) - \hat{\mathsf{p}}_i\hat{\mathsf{p}}_i^T. \tag{1.7}$$

The quantities $(\hat{B}, \hat{p}_i, g_i, H_i)$ can be readily computed from the data $\{(x_i, y_i)\}_{i=1}^n$. To be specific, the MLE $\hat{B}$ in (1.5) can be obtained by invoking a multinomial regression solver (e.g., `sklearn.linear_model.LogisticRegression` from Pedregosa et al. [2011]), and the quantities $\hat{p}_i, g_i, H_i$ can be further computed from eqs. (1.6) and (1.7) by a few matrix multiplications and application of the softmax function.

**Log-odds model and reference class.** The matrix $B^*$ in (1.4) is not identifiable since the conditional distribution of $y_i | x_i$ in the model (1.4) remains unchanged if we replace columns of $B^*$ by $B^* - b\mathbf{1}_{K+1}^T$ for any $b \in \mathbb{R}^p$. In order to obtain an identifiable model, a classical and natural remedy is to model the log-odds, here with the class $K + 1$ as the reference class:

$$\log \frac{\mathbb{P}(y_{ik} = 1 | x_i)}{\mathbb{P}(y_{i(K+1)} = 1 | x_i)} = x_i^T A^* e_k, \qquad \forall k \in [K] \tag{1.8}$$

where $e_k$ is the $k$-th canonical basis vector of $\mathbb{R}^K$, and $A^* \in \mathbb{R}^{p \times K}$ is the unknown parameter. The matrix $A^* \in \mathbb{R}^{p \times K}$ in log-odds model (1.8) is related to $B^* \in \mathbb{R}^{p \times (K+1)}$ in the model (1.4) by $A^* = B^*(I_K, -\mathbf{1}_K)^T$. This log-odds model has two benefits: First it is identifiable since the unknown matrix $A^*$ is uniquely defined. Second, the matrix $A^*$ lends itself well to interpretation as its $k$-th column represents the contrast coefficient between class $k$ and the reference class $K + 1$.

The MLE $\hat{A}$ of $A^*$ in (1.8) is $\hat{A} = \arg\min_{A \in \mathbb{R}^{p \times K}} \sum_{i=1}^n L_i((A, \mathbf{0}_p)^T x_i)$. If the solution set in (1.5) is non-empty, $\hat{A}$ is related to any solution $\hat{B}$ in (1.5) by $\hat{A} = \hat{B}(I_K, -\mathbf{1}_K)^T$. Equivalently,

$$\hat{A}_{jk} = \hat{B}_{jk} - \hat{B}_{j(K+1)} \tag{1.9}$$

for each $j \in [p]$ and $k \in [K]$. If there are three classes (i.e. $K + 1 = 3$), this parametrization allows us to draw scatter plots of realizations of $\sqrt{n} e_j^T \hat{A} = (\sqrt{n}\hat{A}_{j1}, \sqrt{n}\hat{A}_{j2})$ as in Figure 1.

## 1.2 Hypothesis testing for the $j$-th feature and classical asymptotic normality for MLE

**Hypothesis testing for the $j$-th feature.** Our goal is to develop a methodology to test the significance of the $j$-th feature. Specifically, for a desired confidence level $(1 - \alpha) \in (0, 1)$ (say, $1 - \alpha = 0.95$) and a given feature $j \in [p]$ of interest, our goal is to test

$$H_0 : y_i \text{ is conditionally independent of } x_{ij} \text{ given } (x_{ij'})_{j' \in [p] \setminus \{j\}}. \tag{1.10}$$

Namely, we want to test whether the $j$-th variable is independent from the response given all other explanatory variables $(x_{ij'}, j' \in [p] \setminus \{j\})$. Assuming normally distributed $x_i$ and a multinomial model as in (1.4) or (1.8), it is equivalent to test

$$H_0 : e_j^T A^* = \mathbf{0}_K^T \qquad \text{versus} \qquad H_1 : e_j^T A^* \neq \mathbf{0}_K^T, \tag{1.11}$$

where $e_j \in \mathbb{R}^p$ is the $j$-th canonical basis vector.

If the MLE $\hat{B}$ in (1.5) exists in the sense that the solution set in (1.5) is nonempty, the conjecture that rejecting $H_0$ when $e_j^T \hat{B}$ is far from $\mathbf{0}_{K+1}$ is a reasonable starting point. The important question, then, is to determine a quantitative statement for the informal "far from $\mathbf{0}_{K+1}$", similarly to (1.2) or (1.3) in binary logistic regression.

**Classical theory with $p$ fixed.** If $p$ is fixed and $n \to \infty$ in model (1.8), classical maximum likelihood theory [Van der Vaart, 1998, Chapter 5] provides the asymptotic distribution of the MLE $\hat{A}$, which can be further used to test (1.11). Briefly, if $x$ has the same distribution as any $x_i$, the MLE $\hat{A}$ in the multinomial logistic model is asymptotically normal with

$$\sqrt{n}(\text{vec}(\hat{A}) - \text{vec}(A^*)) \xrightarrow{d} N(\mathbf{0}, \mathcal{I}^{-1}) \quad \text{where} \quad \mathcal{I} = \mathbb{E}[(xx^T) \otimes (\text{diag}(\pi^*) - \pi^* \pi^{*T})]$$

is the Fisher information matrix evaluated at the true parameter $A^*$, $\text{vec}(\cdot)$ is the usual vectorization operator, and $\pi^* \in \mathbb{R}^K$ has random entries $\pi_k^* = \exp(x^T A^* e_k) / (1 + \sum_{k'=1}^K \exp(x^T A^* e_{k'}))$ for each $k \in [K]$. In particular, under $H_0 : e_j^T A^* = \mathbf{0}_K^T$,

$$\sqrt{n} \hat{A}^T e_j \xrightarrow{d} N(\mathbf{0}, S_j) \tag{1.12}$$

where $S_j = (e_j^T \otimes I_K)\mathcal{I}^{-1}(e_j \otimes I_K) = e_j^T(\text{cov}(x))^{-1}e_j[\mathbb{E}(\text{diag}(\pi^*) - \pi^*\pi^{*T})]^{-1}$. When (1.12) holds, by the delta method we also have $\sqrt{n}S_j^{-1/2}\hat{A}^Te_j \xrightarrow{\text{d}} N(\mathbf{0}, I_K)$ and

$$n\|S_j^{-1/2}\hat{A}^Te_j\|^2 \xrightarrow{\text{d}} \chi_K^2. \tag{1.13}$$

where the limiting distribution is chi-square with $K$ degrees of freedom. This further suggests the size $\alpha$ test that rejects $H_0$ when $T_n^j(X,Y) > \chi_K^2(\alpha)$, where $T_n^j(X,Y) = n\|S_j^{-1/2}\hat{A}^Te_j\|^2$ is the test statistic. If (1.13) holds, this test is guaranteed to have a type I error converging to $\alpha$. The p-value of this test is given by

$$\int_{T_n^j(X,Y)}^{+\infty} f_{\chi_K^2}(t)dt, \tag{1.14}$$

where $f_{\chi_K^2}(\cdot)$ is the density of the chi-square distribution with $K$ degrees of freedom.

As discussed in the introduction, Sur and Candès [2019] showed that in binary logistic regression, classical normality results for the MLE such as (1.12) fail in the high-dimensional regime because the variance in (1.12) underestimates the variability of the MLE even for null covariates; see also the discussion surrounding (1.2). Our goal is to develop, for classification problems with $K + 1 \geq 3$ classes, a theory that correctly characterize the asymptotic distribution of $\hat{A}^Te_j$ for a null covariate $j \in [p]$ in the high-dimensional regime.

We present first some motivating simulations that demonstrate the failure of classical normal approximation (1.12) in finite samples. These simulations are conducted for various configurations of $(n, p)$ with $K + 1 = 3$ classes. We fix the true parameter $A^*$ and obtain 1000 realizations of $(\hat{A}_{j1}, \hat{A}_{j2})$ by independently resampling the data $\{(x_i, y_i)\}_{i=1}^n$ 1000 times. If the result (1.12) holds, then $\mathbb{P}(\sqrt{n}\hat{A}^Te_j \in \mathcal{C}_\alpha^j) \to 1 - \alpha$, where $\mathcal{C}_\alpha^j = \{u \in \mathbb{R}^K : \|S_j^{-1/2}u\| \leq \chi_K^2(\alpha)\}$. Figure 1 displays scatter plots of $\sqrt{n}(\hat{A}_{j1}, \hat{A}_{j2})$ along with the boundary of 95% confidence set $\mathcal{C}_\alpha^j$ with $\alpha = 0.05$. We observe that, across the three different configurations of $(n, p)$, the 95% confidence sets from our theory (Theorem 2.2 presented in next section) cover around 95% of the realizations, while the set $\mathcal{C}_\alpha^j$ from classical theory only covers approximately 30% of the points, which is significantly lower than the desired coverage rate of 95%. Intuitively and by analogy with results in binary classification [Sur and Candès, 2019], this is because the classical theory (1.12) underestimates the variation of the MLE in the high-dimensional regime. Motivated by this failure of classical MLE theory and the results in binary classification [Sur and Candès, 2019, among others], the goal of this paper is to develop a theory for multinomial logistic regression that achieves the following objectives:

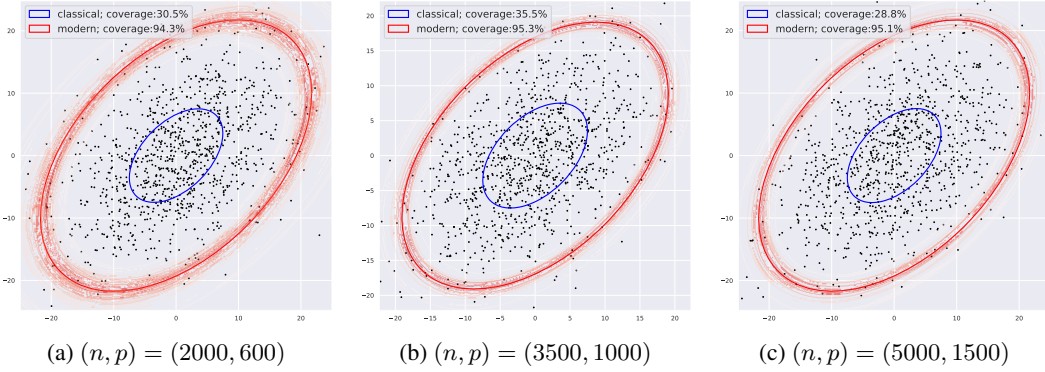

(a) $(n, p) = (2000, 600)$      (b) $(n, p) = (3500, 1000)$      (c) $(n, p) = (5000, 1500)$

Figure 1: Scatter plot of pairs $(\sqrt{n}\hat{A}_{j1}, \sqrt{n}\hat{A}_{j2})$ with $K = 2$ over 1000 repetitions. The blue ellipsoid is the boundary of the 95% confidence set for $\sqrt{n}\hat{A}^Te_j$ under $H_0$ from the classical MLE theory (1.12)-(1.13) based on the Fisher information, the dashed red ellipsoids are the boundaries of the 95% confidence set for $\sqrt{n}\hat{A}^Te_j$ under $H_0$ from this paper (cf. (2.3) below). Each of the 1000 repetition gives a slightly different dashed ellipsoid. The solid red ellipsoid is the average of these 1000 dashed ellipsoids. Each row of $X$ is i.i.d. sampled from $N(\mathbf{0}, \Sigma)$ with $\Sigma = (0.5^{|i-j|})_{p \times p}$. The first $\lceil p/4 \rceil$ rows of $A^*$ are i.i.d. sampled from $N(\mathbf{0}, I_K)$ while other rows are set to zeros. We further normalize $A^*$ such that $A^{*T}\Sigma A^* = I_K$. The last coordinate $j = p$ is used as the null coordinate.

- Establish asymptotic normality of the multinomial MLE $\hat{A}^T e_j$ for null covariates as $n, p \to +\infty$ simultaneously with a finite limit for $n/p$.
- Develop a valid methodology for hypothesis testing of (1.10) in this regime, i.e., testing for the presence of an effect of a feature $j \in [p]$ on the multiclass response.

The contribution of this paper is two-fold: (i) For a null covariate $j \in [p]$, we establish asymptotic normality results for $\hat{A}^T e_j$ that are valid in the high-dimensional regime where $n$ and $p$ have the same order; (ii) we propose a user-friendly test for assessing the significance of a feature in multiclass classification problems.

## 2 Main result: asymptotic normality of $\hat{\mathsf{B}}^T e_j$ and $\hat{A}^T e_j$ on null covariates

In this section, we present the main theoretical results of our work and discuss their significance. We work under the following assumptions.

**Assumption 2.1.** For constants $\delta > 1$, assume that $n, p \to \infty$ with $p/n \leq \delta^{-1}$, and that the design matrix $X \in \mathbb{R}^{n \times p}$ has $n$ i.i.d. rows $(x_i)_{i \in [n]} \sim N(\mathbf{0}, \Sigma)$ for some invertible $\Sigma \in \mathbb{R}^{p \times p}$. The observations $(x_i, \mathsf{y}_i)_{i \in [n]}$ are i.i.d. and each $\mathsf{y}_i$ is of the form $\mathsf{y}_i = f(U_i, x_i^T \mathsf{B}^*)$ for some deterministic function $f$, deterministic matrix $\mathsf{B}^* \in \mathbb{R}^{p \times (K+1)}$ such that $\mathsf{B}^* \mathbf{1}_{K+1} = \mathbf{0}_p$, and latent random variable $U_i$ independent of $x_i$.

**Assumption 2.2** (One-hot encoding). The response matrix $Y$ is in $\mathbb{R}^{n \times (K+1)}$. Its $i$-th row $\mathsf{y}_i$ is a one-hot encoded vector, that is, valued in $\{0, 1\}^{K+1}$ with $\sum_{k=1}^{K+1} \mathsf{y}_{ik} = 1$ for each $i \in [n]$.

The model $\mathsf{y}_i = f(U_i, x_i^T \mathsf{B}^*)$ for some deterministic $f$ and $\mathsf{B}^*$ and latent random variable $U_i$ in Assumption 2.1 is more general than a specific generative model such as the multinomial logistic conditional probabilities in (1.4), as broad choices for $f$ are allowed. In words, the model $\mathsf{y}_i = f(U_i, x_i^T \mathsf{B}^*)$ with $\mathsf{B}^* \mathbf{1}_{K+1} = 0_p$ means that $\mathsf{y}_i$ only depends on $x_i$ through a $K$ dimensional projection of $x_i$ (the projection on the row-space of $\mathsf{B}^*$). The assumption $p/n \leq \delta^{-1}$ is more general than assuming a fixed limit for the ratio $p/n$; this allows us to cover low-dimensional settings satisfying $p/n \to 0$ as well.

The following assumption requires the labels to be "balanced": we observe each class at least $\gamma n$ times for some constant $\gamma > 0$. If $(\mathsf{y}_i)_{i \in [n]}$ are i.i.d. as in Assumption 2.1 with distribution independent of $n, p$, by the law of large numbers this assumption is equivalent to $\min_{k \in [K+1]} \mathbb{P}(\mathsf{y}_{ik} = 1) > 0$.

**Assumption 2.3.** There exits a constant $\gamma \in (0, \frac{1}{K+1}]$, such that for each $k \in [K + 1]$, with probability approaching one at least $\gamma n$ observations $i \in [n]$ are such that $\mathsf{y}_{ik} = 1$. In other words, $\mathbb{P}(\sum_{i=1}^n I(\mathsf{y}_{ik} = 1) \geq \gamma n) \to 1$ for each $k \in [K + 1]$.

As discussed in item list (i) on page 2, in binary logistic regression, Candès and Sur [2020], Sur and Candès [2019] show that the plane $(\frac{p}{n}, \|\Sigma^{1/2}\beta^*\|)$ is split by a smooth curve into two connected open components: in one component the MLE does not exist with high probability, while in the other component, with high probability the MLE exists and is bounded in the sense that $\|\Sigma^{1/2}\hat{\beta}\|^2 < \tau'$ or equivalently $\frac{1}{n}\|X\hat{\beta}\|^2 < \tau$ for constants $\tau, \tau'$ independent of $n, p$. The next assumption requires the typical situation of the latter component, in the current multiclass setting: $\hat{\mathsf{B}}$ in (1.5) exists in the sense that the minimization problem has solutions, and at least one solution is bounded.

**Assumption 2.4.** Assume $\mathbb{P}(\hat{\mathsf{B}}$ exists and $\|X\hat{\mathsf{B}}(I_{K+1} - \frac{\mathbf{1}\mathbf{1}^T}{K+1})\|_F^2 \leq n\tau) \to 1$ as $n, p \to +\infty$ for some large enough constant $\tau$.

Note that the validity of Assumption 2.4 can be assessed using the data at hand; if a multinomial regression solver (e.g. `sklearn.linear_model.LogisticRegression`) converges[1] and $\frac{1}{n}\|X\hat{\mathsf{B}}(I_{K+1} - \frac{\mathbf{1}\mathbf{1}^T}{K+1})\|_F^2$ is no larger than a predetermined large constant $\tau$, then we know Assumption 2.4 holds. Otherwise the algorithm does not converge or produces an unbounded estimate: we know Assumption 2.4 fails to hold and we need collect more data.

---

[1] Here, we refer to standard convergence assessment methods for convex solvers, e.g., looking at the gradient/Hessian values at the current iterate, or looking at the duality gap if available.

Our first main result, Theorem 2.1, provides the asymptotic distribution of $\hat{\mathsf{B}}^T e_j$ where $j \in [p]$ is a null covariate, where $\hat{\mathsf{B}}$ is any minimizer $\hat{\mathsf{B}}$ of (1.5). Throughout, we denote by $\Omega$ the precision matrix defined as $\Omega = \Sigma^{-1}$.

**Theorem 2.1.** *Let Assumptions 2.1 to 2.4 be fulfilled. Then for any $j \in [p]$ such that $H_0$ in (1.10) holds, and any minimizer $\hat{\mathsf{B}}$ of (1.5), we have*

$$\underbrace{\sqrt{\frac{n}{\Omega_{jj}}}}_{scalar} \underbrace{\left(\left(\frac{1}{n}\sum_{i=1}^{n} \mathsf{g}_i \mathsf{g}_i^T\right)^{1/2}\right)^{\dagger}}_{\mathbb{R}^{(K+1)\times(K+1)}} \underbrace{\left(\frac{1}{n}\sum_{i=1}^{n} \mathsf{V}_i\right)}_{\mathbb{R}^{(K+1)\times(K+1)}} \underbrace{\hat{\mathsf{B}}^T e_j}_{\mathbb{R}^{K+1}} \xrightarrow{\mathrm{d}} N\Big(\mathbf{0},\ \underbrace{I_{K+1} - \frac{\mathbf{1}\mathbf{1}^T}{K+1}}_{cov.\ \mathbb{R}^{(K+1)\times(K+1)}}\Big), \quad (2.1)$$

*where $\mathsf{g}_i = -y_i + \hat{\mathsf{p}}_i$ as in (1.7) and $\mathsf{V}_i = \mathsf{H}_i - (\mathsf{H}_i \otimes x_i^T)[\sum_{l=1}^{n} \mathsf{H}_l \otimes (x_l x_l^T)]^{\dagger}(\mathsf{H}_i \otimes x_i)$.*

The proof of Theorem 2.1 is given in Supplementary Section S3. Theorem 2.1 establishes that under $H_0$, $\hat{\mathsf{B}}^T e_j$ converges to a singular multivariate Gaussian distribution in $\mathbb{R}^{K+1}$. In (2.1), the two matrices $\frac{1}{n}\sum_{i=1}^{n}(y_i - \hat{\mathsf{p}}_i)(y_i - \hat{\mathsf{p}}_i)^T$ and $\frac{1}{n}\sum_{i=1}^{n} \mathsf{V}_i$ are symmetric with kernel being the linear span of $\mathbf{1}_{K+1}$, and similarly, if a solution exists, we may replace $\hat{\mathsf{B}}$ by $\hat{\mathsf{B}}(I_{K+1} - \frac{\mathbf{1}\mathbf{1}^T}{K+1})$ which is also solution in (1.5). In this case, all matrix-matrix and matrix-vector multiplications, matrix square root and pseudo-inverse in (2.1) happen with row-space and column space contained in the orthogonal component of $\mathbf{1}_{K+1}$, so that the limiting Gaussian distribution in $\mathbb{R}^{K+1}$ is also supported on this $K$-dimensional subspace.

Since the distribution of the left-hand side of (2.1) is asymptotically pivotal for all null covariates $j \in [p]$, Theorem 2.1 opens the door of statistical inference for multinomial logistic regression in high-dimensional settings. By construction, the multinomial logistic estimate $\hat{A} \in \mathbb{R}^{p \times K}$ in (1.9) ensures $(\hat{A}, \mathbf{0}_p)$ is a minimizer of (1.5). Therefore, we can deduce the following theorem from Theorem 2.1.

**Theorem 2.2.** *Define the matrix $R = (I_K, \mathbf{0}_K)^T \in \mathbb{R}^{(K+1)\times K}$ using block matrix notation. Let Assumptions 2.1 to 2.4 be fulfilled. For $\hat{A}$ in (1.9) and any $j \in [p]$ such that $H_0$ in (1.10) holds,*

$$\underbrace{\left(I_K + \frac{\mathbf{1}_K \mathbf{1}_K^T}{\sqrt{K+1}+1}\right)R^T}_{matrix\ \mathbb{R}^{K\times(K+1)}} \underbrace{\sqrt{\frac{n}{\Omega_{jj}}}}_{scalar} \underbrace{\left(\left(\frac{1}{n}\sum_{i=1}^{n}\mathsf{g}_i\mathsf{g}_i^T\right)^{1/2}\right)^{\dagger}}_{matrix\ \mathbb{R}^{(K+1)\times(K+1)}} \underbrace{\left(\frac{1}{n}\sum_{i=1}^{n}\mathsf{V}_i R\right)}_{\mathbb{R}^{(K+1)\times K}} \underbrace{\hat{A}^T e_j}_{\mathbb{R}^K} \xrightarrow{\mathrm{d}} N(\mathbf{0}_K, I_K) \quad (2.2)$$

*where $\mathsf{g}_i$ is defined in (1.7) and $\mathsf{V}_i$ is defined in Theorem 2.1. Furthermore, for the same $j \in [p]$,*

$$\mathcal{T}_n^j(X,Y) \stackrel{def}{:=} \frac{n}{\Omega_{jj}}\left\|\left(\left(\frac{1}{n}\sum_{i=1}^{n}\mathsf{g}_i\mathsf{g}_i^T\right)^{1/2}\right)^{\dagger}\left(\frac{1}{n}\sum_{i=1}^{n}\mathsf{V}_i\right)R\hat{A}^T e_j\right\|^2 \text{ satisfies } \mathcal{T}_n^j(X,Y) \xrightarrow{\mathrm{d}} \chi_K^2. \quad (2.3)$$

*Note that Equations (2.2) and (2.3) is stated using $\Omega_{jj} = e_j^T \Sigma^{-1} e_j$. When $\Sigma$ is unknown, the quantity $\Omega_{jj}$ in above results can be replaced by its consistent estimate $\hat{\Omega}_{jj}$ defined in (2.5), and the convergence in distribution results still hold.*

Theorem 2.2 is proved in Supplementary Section S4. To the best of our knowledge, Theorem 2.2 is the first result that characterizes the distribution of null MLE coordinate $\hat{A}^T e_j$ in high-dimensional multinomial logistic regression with 3 or more classes. It is worth mentioning that the quantities $(\mathsf{g}_i, \mathsf{V}_i, \hat{A})$ used in Theorem 2.2 can be readily computed from the data $(X, Y)$. Therefore, Theorem 2.2 lets us test the significance of a specific feature: for testing $H_0$, this theorem suggests the test statistic $\mathcal{T}_n^j(X, Y)$ in (2.3) and the rejection region $\mathcal{E}_\alpha^j \stackrel{def}{:=} \{(X, Y) : \mathcal{T}_n^j(X, Y) \geq \chi_K^2(\alpha)\}$. Under the null hypothesis $H_0$ in (1.10), Theorem 2.2 guarantees $\mathbb{P}\big((X, Y) \in \mathcal{E}_\alpha^j\big) \to \alpha$. In other words, the test that rejects $H_0$ if $(X, Y) \in \mathcal{E}_\alpha^j$ has type I error converging to $\alpha$. The p-value of this test is

$$\text{p-value} = \int_{\mathcal{T}_n^j(X,Y)}^{+\infty} f_{\chi_K^2}(t)dt, \quad (2.4)$$

where $f_{\chi_K^2}(\cdot)$ is the density of the chi-square distribution with $K$ degrees of freedom.

**Unknown $\Omega_{jj} = e_j^T \Sigma^{-1} e_j$.** If $\Sigma$ is unknown, we describe a consistent estimate of the quantity $\Omega_{jj}$ appearing in (2.1), (2.2), and (2.3). Under the Gaussian Assumption 2.1, the quantity $\Omega_{jj}$ is the

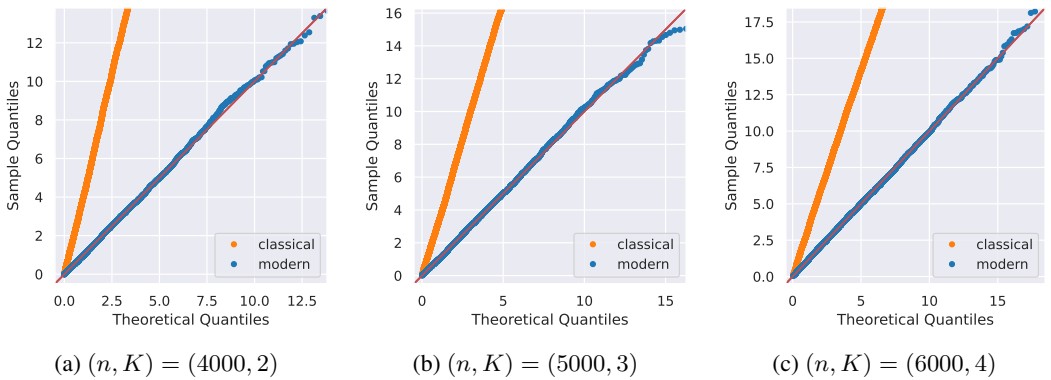

(a) $(n, K) = (4000, 2)$  (b) $(n, K) = (5000, 3)$  (c) $(n, K) = (6000, 4)$

Figure 2: Q-Q plots of the test statistic in the left-hand side of (1.13) (in orange) and in the left-hand side of (2.3) (in blue) for different $(n, K)$ and $p = 1000$.

reciprocal of the conditional variance $\mathrm{Var}(x_{ij}|x_{i,-j})$, which is also the noise variance in the linear model of regressing $Xe_j$ onto $X_{-j}$ (the submatrix of $X$ excluding the $j$-th column). According to standard results in linear models, we have $\Omega_{jj}\|[I_n - X_{-j}(X_{-j}^T X_{-j})^{-1}X_{-j}^T]Xe_j\|^2 \sim \chi^2_{n-p+1}$. Since $\chi^2_{n-p+1}/(n-p+1) \to 1$ almost surely by the strong law of large numbers,

$$\hat{\Omega}_{jj} = (n - p + 1)/\|[I_n - X_{-j}(X_{-j}^T X_{-j})^{-1}X_{-j}^T]Xe_j\|^2 \tag{2.5}$$

is a consistent estimator of $\Omega_{jj}$. Therefore, the previous asymptotic results in Theorems 2.1 and 2.2 still hold by Slutsky's theorem if we replace $\Omega_{jj}$ by the estimate $\hat{\Omega}_{jj}$ in (2.5).

## 3  Numerical experiments

This section presents simulations and a real data analysis to examine finite sample properties of the above results and methods. The source code for generating all of the experimental results in this paper can be found in the supplementary material.

**Simulation settings.** We set $p = 1000$ and consider different combinations of $(n, K)$. The covariance matrix $\Sigma$ is specified to be the correlation matrix of an AR(1) model with parameter $\rho = 0.5$, that is, $\Sigma = (0.5^{|i-j|})_{p \times p}$. We generate the regression coefficients $A^* \in \mathbb{R}^{p \times K}$ once and for all as follows: sample $A_0 \in \mathbb{R}^{p \times K}$ with first $\lceil p/4 \rceil$ rows being i.i.d. $N(\mathbf{0}, I_K)$, and set the remaining rows to 0. We then scale the coefficients by defining $A^* = A_0(A_0^T \Sigma A_0)^{-1/2}$ so that $A^{*T}\Sigma A^* = I_K$. With this construction, the $p$-th variable is always a null covariate , and we use this null coordinate $j = p$ to demonstrate the effectiveness of our theoretical results presented in Theorem 2.2 and the suggested test for testing $H_0$ as described in (1.10). Using the above settings, we generate the design matrix $X \in \mathbb{R}^{n \times p}$ from $N(0, \Sigma)$, and then simulate the labels from a multinomial logistic model as given in (1.8), using the coefficients $A^* \in \mathbb{R}^{p \times K}$. For each simulation setting, we perform 5,000 repetitions.

**Assessment of $\chi^2$ approximations.** To assess the $\chi^2$ approximation (2.3) from this paper and that of the classical theory (1.13), we compute the two $\chi^2_K$ test statistics for each sample $(x_i, y_i)_{i=1}^n$. Figure 2 shows the empirical quantiles of the two statistics versus the $\chi^2_K$ distribution quantiles. The results demonstrate that the quantiles (in blue) from our high-dimensional theory closely match the 45-degree line (in red), whereas the quantiles (in orange) from the classical theory significantly deviate from the 45-degree line. These findings highlight the accuracy of our proposed $\chi^2$ approximation (2.3) over the classical result (1.13) when $p$ is not sufficiently small compared to $n$.

**Uniformity of null p-values.** Recall that the p-value from the classical test (1.13) is given by (1.14), while the p-value from this paper taking into account high-dimensionality is given by (2.4). Figure 3 displays the histograms of these two sets of p-values out of 5000 repetitions. The results in Figure 3 show that the p-values obtained from the classical test deviate significantly from the uniform distribution, with a severe inflation in the lower tail. This indicates that the classical test tends to produce large type I errors due to the excess of p-values close to 0. In contrast, the p-values proposed in this paper exhibit a uniform distribution, further confirming the effectiveness and applicability of the theory in Theorem 2.2 for controlling type I error when testing for null covariates with (1.10).

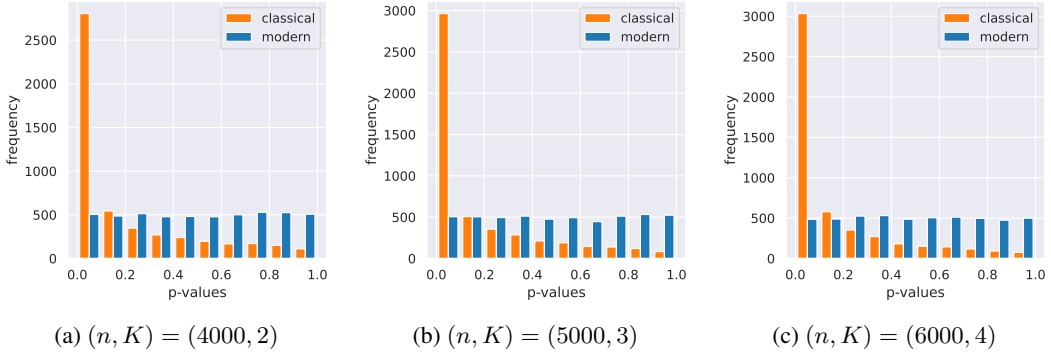

Figure 3: Histogram for p-values of the classical test (1.14) (in orange) and of the proposed test (2.4) (in blue) under $H_0$ in simulated data with different $(n, K)$ and $p = 1000$.

**Unknown $\Omega_{jj}$.** In the situation where the covariance matrix $\Sigma$ is unknown, we can estimate the diagonal element $\Omega_{jj} = e_j^T \Sigma^{-1} e_j$ by $\hat{\Omega}_{jj}$ defined in (2.5). To evaluate the accuracy of the normal and chi-square approximations and the associated test with $\Omega_{jj}$ replaced by $\hat{\Omega}_{jj}$, we conduct simulations similar to those in Figures 2 and 3, but we replace $\Omega_{jj}$ with its estimate $\hat{\Omega}_{jj}$. The results are presented in Figure S1. The plots are visually indistinguishable from the plots using $\Omega_{jj}$. These confirm that the chi-square approximation and the associated test using $\hat{\Omega}_{jj}$ are accurate.

**Non-Gaussian covariates and unknown $\Omega_{jj}$.** Although our theory assumes Gaussian covariates, we expect that the same results hold for other distributions with sufficiently light tails. To illustrate this point, we consider the following two types of non-Gaussian covariates: (i) The design matrix $X$ has i.i.d. Rademacher entries, i.e., $\mathbb{P}(x_{ij} = \pm 1) = \frac{1}{2}$, (ii) Each $x_{ij}$ takes on values 0, 1 and 2 with respectively probabilities $a_j^2, 2a_j(1 - a_j)$, and $(1 - a_j)^2$, where $a_j$ varies in $[0.25, 0.75]$. Each columns of $X$ are then centered and normalized to have 0 mean and unit variance. This generation of non-Gaussian covariates is adopted from single-nucleotide poly-morphisms (SNPs) example in Sur and Candès [2019]. For these two types of non-Gaussian covariates, we further rescale the feature vectors to ensure that $x_i$ has the same covariance as in the Gaussian case at the beginning of Section 3, that is $\Sigma = (0.5^{|i-j|})_{p \times p}$. We present the Q-Q plots in Figure S2 using the same settings as in Figure S1, with the only difference being that the covariates in Figure S2 are non-Gaussian distributed. The Q-Q plots of $\mathcal{T}_n^j(X, Y)$ in (2.3) plotted in Figure S2 still closely match the diagonal line. These empirical successes suggest that the normal and $\chi_K^2$ approximations (2.1)-(2.3) apply to a wider range of covariate distributions beyond normally distributed data.

**Real data example.** We conduct a real data analysis by applying the proposed test to heart disease data from the UCI Machine Learning Repository (link: http://archive.ics.uci.edu/ml/machine-learning-databases/heart-disease/processed.cleveland.data). After standard data-cleaning processes, the dataset has 297 instances with 13 features, including age, sex, and other attributes. The response variable was transformed into 3 classes (0, 1, and 2) after converting the labels 3 and 4 to 2. To demonstrate the validity of the proposed significance test, we generate a noise variable from a standard normal distribution, resulting in a dataset with 297 instances and 14 variables. We test the significance of this noise variable using both the proposed chi-square test and the classical test, repeating the experiment 10,000 times. The type I error of our proposed test is 0.0508, aligning well with the desired type I error of 0.05. In contrast, the classical test exhibits a type I error of 0.0734, significantly exceeding the desired rate of 0.05. These results confirm the validity of our proposed test on real data corrupted with fake covariate, while the classical test concludes that the fake covariate is significant in more than 7% of experiments, leading to false discoveries exceeding the desired 0.05 type I error.

## 4 Discussion and future work

Multinomial logistic regression estimates and their p-values are ubiquitous throughout the sciences for analyzing the significance of explanatory variables on multiclass responses. Following the seminal work of Sur and Candès [2019] in binary logistic regression, this paper develops the first valid tests and p-values for multinomial logistic estimates when $p$ and $n$ are of the same order. For 3 or more

classes, this methodology and the corresponding asymptotic normality results in Theorems 2.1 and 2.2 are novel and provide new understanding of multinomial logistic estimates (also known as cross-entropy minimizers) in high-dimensions. We expect similar asymptotic normality and chi-square results to be within reach for loss functions different than the cross-entropy or a different model for the response $y_i$; for instance Section S1 provides an extension to the $q$-repeated measurements model, where $q$ responses are observed for each feature vector $x_i$.

Let us point a few follow-up research directions that we leave open for future work. A first open problem regards extensions of our methodology to confidence sets for $e_j^T B^*$ when $H_0$ in (1.10) is violated for the $j$-th covariate. This would require more stringent assumptions on the generative model than Assumption 2.1 as $B^*$ there is not identifiable (e.g., modification of both $B^*$ and $f(\cdot, \cdot)$ in Assumption 2.1 is possible without changing $y_i$). A second open problem is to relate this paper's theory to the fixed-point equations and limiting Gaussian model obtained in multiclass models, e.g., Loureiro et al. [2021]. While it may be straightforward to obtain the limit of $\frac{1}{n}\sum_{i=1}^{n} g_i g_i^T$ and of the empirical distribution of the rows of $\hat{A}$ in this context (e.g., using Corollary 3 in Loureiro et al. [2021]), the relationship between the fixed-point equations and the matrix $\frac{1}{n}\sum_{i=1}^{n} V_i$ appearing in (2.1) is unclear and not explained by typical results from this literature. A third open problem is to characterize the exact phase transition below which the multinomial logistic MLE exists and is bounded with high-probability (Assumption 2.4); while this is settled for two classes [Candès and Sur, 2020] and preliminary results are available for 3 or more classes [Loureiro et al., 2021, Kini and Thrampoulidis, 2021], a complete understanding of this phase transition is currently lacking. A last interesting open problem is to prove that our theory extend to non-Gaussian data, as observed in simulations. This challenging problem is often referred to as "universality" and has received intense attention recently [Gerace et al., 2022, Han and Shen, 2022, Montanari and Saeed, 2022, Pesce et al., 2023, Dandi et al., 2023], showing that in several settings of interest (although none exactly matching the one considered in the present paper), the asymptotic behavior of the minimizers is unchanged if the distribution of the covariates is modified from normal to another distribution with the same covariance.

## Acknowledgments and Disclosure of Funding

P.C. Bellec's research was partially supported by the NSF Grant DMS-1945428. The authors acknowledge the Office of Advanced Research Computing (OARC) at Rutgers, The State University of New Jersey for providing access to the Amarel cluster and associated research computing resources that have contributed to the results reported here. URL: https://oarc.rutgers.edu.

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

# Supplementary Material of "Multinomial Logistic Regression: Asymptotic Normality on Null Covariates in High-Dimensions"

Let us define some standard notation that will be used in the rest of this supplement. For a vector $v \in \mathbb{R}^n$, let $\|v\|_\infty = \max_{i \in [n]} |v_i|$ denote the infinity norm of vector $v$. If $A$ is symmetric, we define $\lambda_{\min}(A)$ and $\|A\|_{op}$ as the minimal and maximal eigenvalues of $A$, respectively. For two symmetric matrices $A, B$ of the same size, we write $A \preceq B$ if and only if $B - A$ is positive semi-definite.

## Diagram: Organization of the proofs

The following diagram summarizes the different theorems and lemmas, and the relationships between them.

**Theorem 2.2**

Asymptotic normality for $\hat{A}^T e_j$ on null covariates, where $\hat{A} \in \mathbb{R}^{p \times K}$ is the multinomial logistic MLE with class $K + 1$ fixed as the reference class (see (1.9)).

**Theorem 2.1**

Asymptotic normality for $\hat{B}^T e_j$ on null covariates, where $\hat{B} \in \mathbb{R}^{p \times (K+1)}$ is the multinomial logistic MLE in (1.5).

**Theorem S3.1**

Asymptotic normality for $\hat{B}^T e_j$ on null covariates, where $\hat{B} \in \mathbb{R}^{p \times K}$ is the multinomial logistic MLE using the parameter space from Section S3.1. The proof uses that the conditions in Theorem S5.1 on the loss function are satisfied by the cross-entropy.

**$K$-dimensional orthogonal parametrization defined by the matrix the $Q$**

Section S3.1 defines the matrix $Q \in \mathbb{R}^{(K+1) \times K}$ and discusses a convenient parametrization of the model isometric to the subspace orthogonal to $\mathbf{1}_{K+1}$.

**Control of $\mathsf{g}_i$ and $\mathsf{H}_i$ for the cross-entropy loss**

Lemmas S6.1 and S6.2 give deterministic arguments to control the gradients and Hessians of the cross-entropy loss. Lemma S6.4 controls the Hessian of the cross-entropy loss at the minimizer, in a specific high-probability event. Lemma S6.3 defines this high-probability event.

**Theorem S5.1**

Asymptotic normality on null covariates for general loss functions, $\Sigma \neq I_p$. Deduced from Theorem S5.2 by rotational invariance.

**Lemma S5.5**

Lemma S5.5 computes the derivatives of the minimizer with respect to $X$, used in the proof of Theorem S5.2.

**Theorem S5.2**

Asymptotic normality on null covariates for general loss functions, $\Sigma = I_p$.

**Lemma S5.3**

Normal and $\chi^2$ approximations for random variables defined as a differentiable function of standard normal vectors.

## S1 Extension: $q$ repeated measurements

Let integer $q \geq 1$ be a constant independent of $n, p$. Our results readily extend if $q$ labels are observed for each observed feature vector $x_i$, and the corresponding $q$ one-hot encoded vectors are averaged into $\mathsf{y}_i \in \{0, \frac{1}{q}, \frac{2}{q}, ..., 1\}^{K+1}$. Concretely, for each observation $i \in [n]$, $q$ i.i.d. labels $(Y_i^m)_{m \in [q]}$ are observed with each $Y_i^m \in \{0, 1\}^{K+1}$ one-hot encoded and $\mathsf{y}_{ik} = \frac{1}{q} \sum_{m=1}^q Y_{ik}^m$, for instance in a repeated multinomial regression model with $\mathbb{P}(Y_{ik}^m = 1 | x_i)$ equal to right-hand side of (1.4). In this case where $(Y_i^m)_{m \in [q]}$ are i.i.d., Assumption 2.3 is satisfied by the law of large numbers if $\min_{k \in [K+1]} \mathbb{P}(Y_{ik}^m = 1) > 0$ since $q$ is constant. For this $q$ repeated measurements model, the negative log-likelihood function of a parameter $\mathsf{B} \in \mathbb{R}^{p \times (K+1)}$ is

$$- \sum_{i=1}^n \sum_{m=1}^q \sum_{k=1}^{K+1} Y_{ik}^m \Big[ x_i^T \mathsf{B} e_k - \log \sum_{k'=1}^{K+1} \exp(x_i^T \mathsf{B} e_{k'}) \Big]$$

$$= q \sum_{i=1}^n \sum_{k=1}^{K+1} \mathsf{y}_{ik} \Big[ -x_i^T \mathsf{B} e_k + \log \sum_{k'=1}^{K+1} \exp(x_i^T \mathsf{B} e_{k'}) \Big]$$

$$= q \sum_{i=1}^n \Big[ \sum_{k=1}^{K+1} -\mathsf{y}_{ik} x_i^T \mathsf{B} e_k + \log \sum_{k'=1}^{K+1} \exp(x_i^T \mathsf{B} e_{k'}) \Big]$$

$$= q \sum_{i=1}^n \mathsf{L}_i(\mathsf{B}^T x_i),$$

where the first equality uses $\mathsf{y}_{ik} = \frac{1}{q} \sum_{m=1}^q Y_{ik}^m$, the second equality uses $\sum_{k=1}^{K+1} \mathsf{y}_{ik} = 1$ under the following Assumption S1.1, and the last equality uses the definition of $\mathsf{L}_i$ after (1.5).

**Assumption S1.1.** For all $i \in [n]$, the response $\mathsf{y}_i$ is in $\{0, 1/q, 2/q, ..., 1\}^{K+1}$ with $\sum_{k=1}^{K+1} \mathsf{y}_{ik} = 1$.

In such repeated measurements model, we replace Assumption 2.2 with Assumption S1.1 under which the following Theorem S1.1 holds.

**Theorem S1.1.** *Let $q \geq 2$ be constant. Let Assumptions S1.1, 2.1, 2.3 and 2.4 be fulfilled. For any $j \in [p]$ such that $H_0$ in (1.10) holds, we have the convergence in distribution (2.1), (2.2) and (2.3).*

*Proof of Theorem S1.1.* Under the assumptions in Theorem S1.1, the MLE $\hat{\mathsf{B}}$ for this $q$ repeated measurements model is the minimizer of the optimization problem

$$\hat{\mathsf{B}} \in \underset{\mathsf{B} \in \mathbb{R}^{p \times (K+1)}}{\arg\min} \sum_{i=1}^n \mathsf{L}_i(\mathsf{B}^T x_i)$$

as in (1.5). Similar to the non-repeated model, the MLE $\hat{A}$ for the identifiable log-odds model can be expressed as

$$\hat{A} = \underset{A \in \mathbb{R}^{p \times K}}{\arg\min} \sum_{i=1}^n \mathsf{L}_i((A, \mathbf{0}_p)^T x_i).$$

The only difference between this $q$ repeated measurements model and the non-repeated model considered in the main text is that the response $\mathsf{y}_{ik}$ for this $q$ repeated measurements model is now valued in $\{0, 1/q, 2/q, ..., 1\}$. Because the proofs of Theorems 2.1 and 2.2 do not require the value of $\mathsf{y}_{ik}$ to be $\{0, 1\}$-valued. Theorem S1.1 can be proved by the same arguments used in the proof of Theorems 2.1 and 2.2. $\qquad\square$

## S2 Implementation details and additional figures

The pivotal quantities in our main results Theorems 2.1 and 2.2 involve only observable quantities that can be computed from the data $(x_i, \mathsf{y}_i)_{i \in [n]}$. In this section we provide an efficient way of computing the matrix $\mathsf{V}_i$ appearing in Theorems 2.1 and 2.2.

**Fast computation of $\mathsf{V}_i$.** Recall the definition of $\mathsf{V}_i$ in Theorem 2.1,

$$\mathsf{V}_i = \mathsf{H}_i - (\mathsf{H}_i \otimes x_i^T)\Big[\sum_{l=1}^{n}\mathsf{H}_l \otimes (x_l x_l^T)\Big]^\dagger (\mathsf{H}_i \otimes x_i).$$

The majority of computational cost in calculating $\mathsf{V}_i$ lies in the step of calculating its second term

$$(\mathsf{H}_i \otimes x_i^T)\Big[\sum_{l=1}^{n}\mathsf{H}_l \otimes (x_l x_l^T)\Big]^\dagger (\mathsf{H}_i \otimes x_i^T).$$

Here we provide an efficient way to compute this term using the Woodbury matrix identity. Since $\mathsf{H}_i \mathbf{1}_{K+1} = \mathbf{0}_{K+1}$, we have $\ker(\mathsf{H}_i \otimes (x_i x_i^T))$ is the span of $\{\mathbf{1}_{K+1} \otimes e_j : j \in [p]\}$, where $\mathbf{1}_{K+1}$ is the all-ones vector in $\mathbb{R}^{K+1}$. Therefore, the second term in $\mathsf{V}_i$ can be rewritten as

$$(\mathsf{H}_i \otimes x_i^T)\Big[\sum_{l=1}^{n}\mathsf{H}_l \otimes (x_l x_l^T)\Big]^\dagger (\mathsf{H}_i \otimes x_i)$$

$$= (\mathsf{H}_i \otimes x_i^T)\Big[\sum_{l=1}^{n}\mathsf{H}_l \otimes (x_l x_l^T) - \sum_{j=1}^{p}(\mathbf{1}\otimes e_j)(\mathbf{1}\otimes e_j)^T\Big]^{-1}(\mathsf{H}_i \otimes x_i).$$

We now apply the Woodbury matrix identity to compute the matrix inversion in the above display. Recall $\mathsf{H}_i = \mathrm{diag}(\hat{\mathsf{p}}_i) - \hat{\mathsf{p}}_i \hat{\mathsf{p}}_i^T$, we have

$$\sum_{i=1}^{n}\mathsf{H}_i \otimes (x_i x_i^T) = \sum_{k=1}^{K+1}(e_k e_k^T)\otimes\big(\sum_{i=1}^{n}\hat{\mathsf{p}}_{ik}x_i x_i^T\big) - \sum_{i=1}^{n}(\hat{\mathsf{p}}_i \otimes x_i)(\hat{\mathsf{p}}_i \otimes x_i)^T.$$

Let $A = \sum_{k=1}^{K+1}(e_k e_k^T)\otimes(\sum_{i=1}^{n}\hat{\mathsf{p}}_{ik}x_i x_i^T)$, and $U \in \mathbb{R}^{p(K+1)\times(n+p)}$ with the first $n$ columns being $(\hat{\mathsf{p}}_i \otimes x_i)_{i\in[n]}$ and the following $p$ columns $(\mathbf{1}\otimes e_j)_{j\in[p]}$. Then the term we want to invert is $A - UU^T$, where $A$ is a block diagonal matrix and can be inverted by inverting each block separately. By the Woodbury matrix identity, we have

$$(A - UU^T)^{-1} = A^{-1} - A^{-1}U(-I_{n+p} + U^T A^{-1}U)^{-1}U^T A^{-1}.$$

The gain of using the above formula is significant for large $K$: instead of inverting the $p(K+1)\times p(K+1)$ matrix $\sum_{l=1}^{n}\mathsf{H}_l \otimes (x_l x_l^T)$ in the left-hand side, the right-hand side only requires to invert a block diagonal matrix $A$ and a $(n+p)\times(n+p)$ matrix $-I_{n+p} + U^T A^{-1}U$.

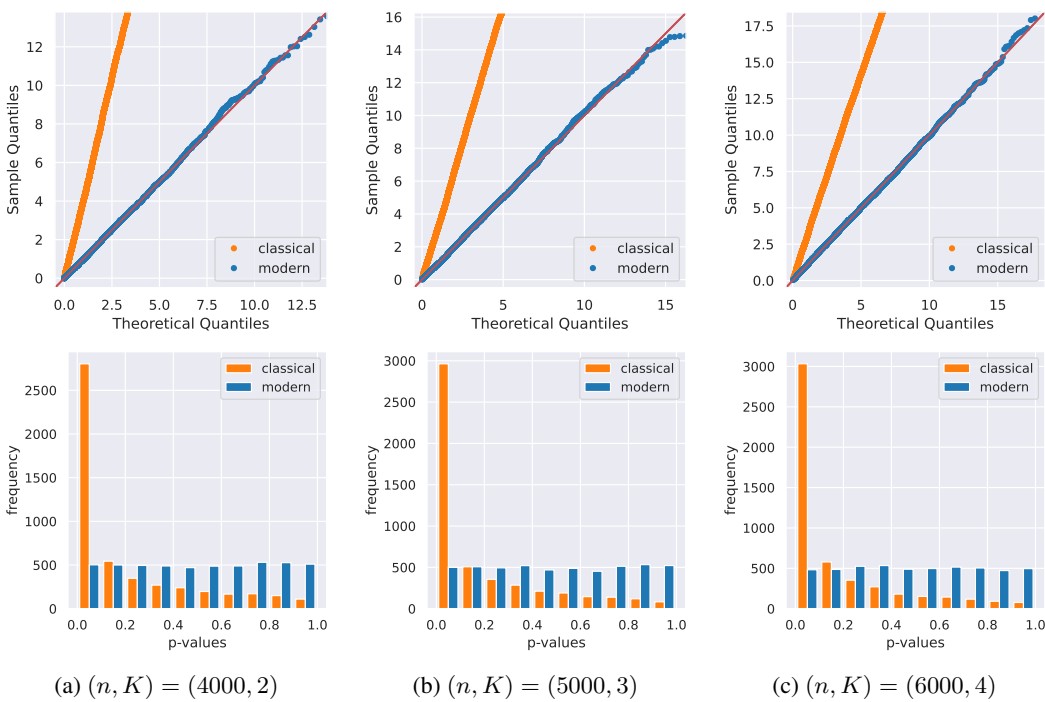

Figure S1: **The upper row:** Q-Q plots of the test statistics from (2.3) (in blue) and (1.13) (in orange) for different $(n, K)$ and $p = 1000$ using $\hat{\Omega}_{jj}$. **The lower row:** histograms of p-values from classical test and our test for different $(n, K)$ and $p = 1000$ using $\hat{\Omega}_{jj}$.

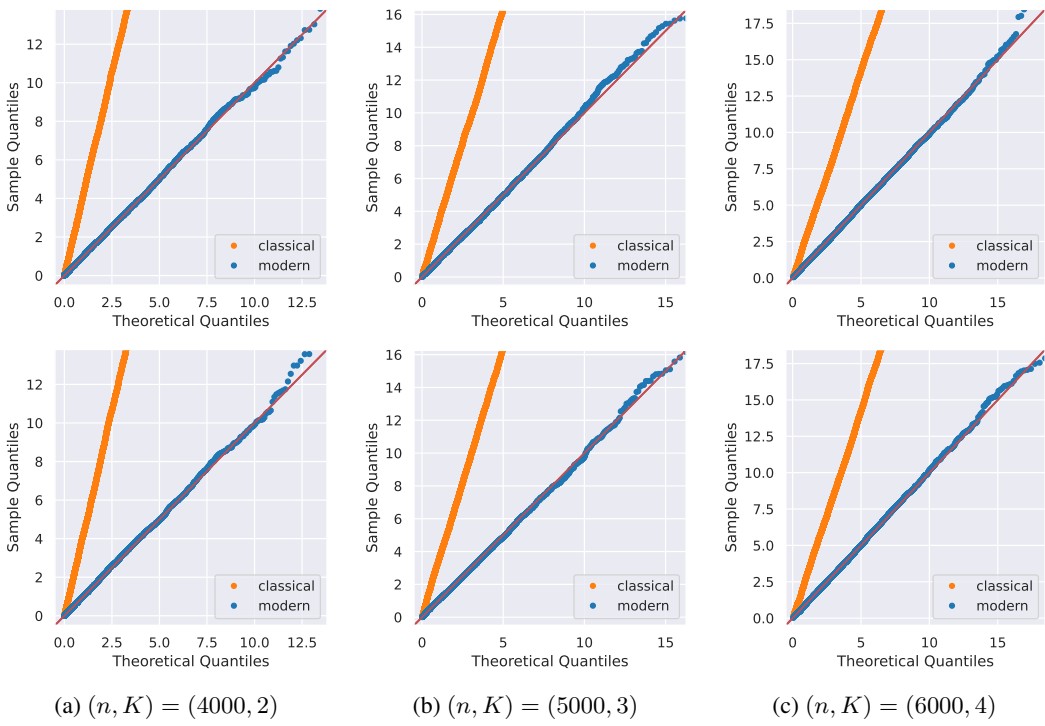

Figure S2: Q-Q plots of the test statistics from (2.3) (in blue) and (1.13) (in orange) for different $(n, K)$ and $p = 1000$ using $\hat{\Omega}_{jj}$. **The upper row:** covariates are sampled from Rademacher distribution. **The lower row:** covariates are sampled from distribution of SNPs.

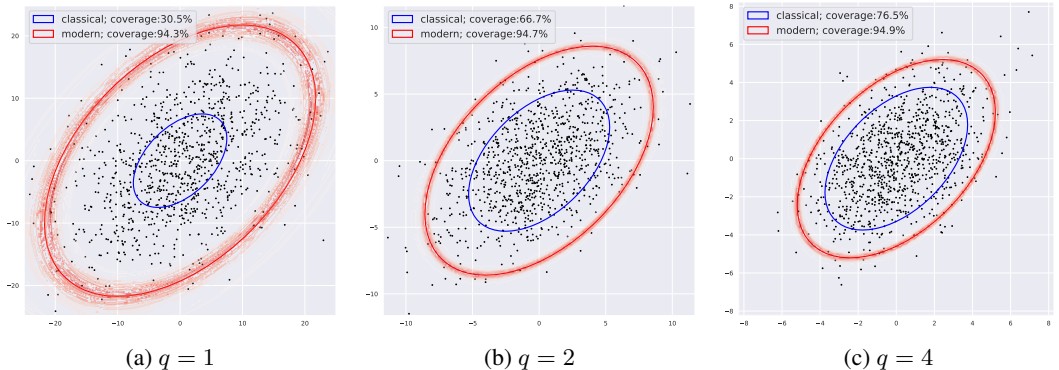

(a) $q = 1$          (b) $q = 2$          (c) $q = 4$

Figure S3: Scatter plot of pairs $(\sqrt{n}\hat{A}_{j1}, \sqrt{n}\hat{A}_{j2})$ with the same data generating process as in Figure 1 (a) except using different $q$.

## S3   Proof of Theorem 2.1

Before proving Theorem 2.1, we present another parametrization of the multinomial logistic regression model. The asymptotic theory of MLE for this new parametrized multinomial logistic model will be used to prove Theorem 2.1.

### S3.1   Another parametrization of multinomial logistic regression

Recall the symbol "$\in$" is used in (1.5) to emphasize that the minimizer $\hat{\mathsf{B}}$ in (1.5) is not unique: if $\hat{\mathsf{B}}$ is a minimizer of (1.5) then $\hat{\mathsf{B}} - b\mathbf{1}_{K+1}^T$ is also a minimizer of (1.5), for any $b \in \mathbb{R}^p$ and the all-ones vector $\mathbf{1}_{K+1}$ in $\mathbb{R}^{K+1}$. Besides the log-odds model (1.8), here we consider another identifiable parametrization of multinomial logistic regression, whose unknown parameter, denoted by $B^*$, is in $\mathbb{R}^{p \times K}$.

**Orthogonal complement.** To obtain an identifiable multinomial logistic regression model from (1.4), we consider the symmetric constraint $\mathsf{B}^* \mathbf{1} = \sum_{k=1}^{K+1} \mathsf{B}^* e_k = \mathbf{0}$ as in [Zhu and Hastie, 2004], thus $\mathsf{B}^* = \mathsf{B}^*(I_{K+1} - \frac{\mathbf{1}\mathbf{1}^T}{K+1})$, where $\mathbf{1}$ is the all-ones vector in $\mathbb{R}^{K+1}$. Let $Q \in \mathbb{R}^{(K+1)\times K}$ be any matrix such that

$$I_{K+1} - \frac{1}{K+1}\mathbf{1}\mathbf{1}^T = QQ^T, \qquad Q^T Q = I_K. \tag{S3.1}$$

We fix one choice of $Q$ satisfying (S3.1) throughout this supplement. Let $B^* = \mathsf{B}^* Q$, then $\mathsf{B}^* = B^* Q^T$ and the model (1.4) can be parameterized using $B^*$ as

$$\mathbb{P}(\mathsf{y}_{ik} = 1|x_i) = \frac{\exp(x_i^T B^* Q e_k)}{\sum_{k'=1}^{K+1} \exp(x_i^T B^* Q e_{k'})}, \quad k \in \{1, 2, \dots, K+1\}. \tag{S3.2}$$

The multinomial logistic MLE of $B^*$ in (S3.2) is given by

$$\hat{B} = \arg\min_{B \in \mathbb{R}^{p \times K}} \sum_{i=1}^{n} L_i(B^T x_i), \tag{S3.3}$$

where $L_i : \mathbb{R}^K \to \mathbb{R}$ is defined by $L_i(u) = \mathsf{L}_i(Qu)$ for all $u \in \mathbb{R}^K$. By this construction, we have $\hat{B} = \hat{\mathsf{B}}Q$ for any minimizer $\hat{\mathsf{B}}$ of (1.5). Furthermore, by the chain rule using the expressions (1.7), the gradient and Hessian of $L_i$ evaluated at $\hat{B}^T x_i$ are

$$g_i := \nabla L_i(\hat{B}^T x_i) = Q^T \mathsf{g}_i, \qquad H_i := \nabla^2 L_i(\hat{B}^T x_i) = Q^T \mathsf{H}_i Q. \tag{S3.4}$$

Throughout, we use serif upright letters to denote quantities defined on the unidentifiable parameter space $\mathbb{R}^{p \times (K+1)}$:

$$\mathsf{B}^*, \hat{\mathsf{B}} \in \mathbb{R}^{p \times (K+1)}, \quad \mathsf{L}_i : \mathbb{R}^{K+1} \to \mathbb{R}, \qquad \hat{\mathsf{p}}_i, \mathsf{y}_i, \mathsf{g}_i \in \mathbb{R}^{K+1}, \qquad \mathsf{H}_i \in \mathbb{R}^{(K+1)\times(K+1)}$$

and the normal italic font to denote analogous quantities for the identifiable parameter space $\mathbb{R}^{p \times K}$:

$$B^*, \hat{B} \in \mathbb{R}^{p \times K}, \qquad L_i : \mathbb{R}^K \to \mathbb{R}, \qquad g_i \in \mathbb{R}^K, \qquad H_i \in \mathbb{R}^{K \times K}.$$

Theorem S3.1 provides the asymptotic normality and the chi-square approximation of null MLE coordinates in high-dimensions where $n, p \to \infty$ with the ratio $n/p$ converging to a finite limit.

**Theorem S3.1** (Proof is given on page 27). *Let Assumptions 2.1, 2.3 and 2.4 be fulfilled. Assume that either Assumption 2.2 or Assumption S1.1 holds. Then for any $j \in [p]$ such that $H_0$ in (1.10) holds,*

$$\sqrt{n}\Omega_{jj}^{-1/2}\Big(\frac{1}{n}\sum_{i=1}^{n}g_ig_i^T\Big)^{-1/2}\Big(\frac{1}{n}\sum_{i=1}^{n}V_i\Big)\hat{B}^Te_j \xrightarrow{\text{d}} N(0, I_K), \qquad (S3.5)$$

*where $V_i = H_i - (H_i \otimes x_i^T)[\sum_{l=1}^{n}H_l \otimes (x_lx_l^T)]^{-1}(H_i \otimes x_i)$.*

*A direct consequence of* (S3.5) *is the $\chi^2$ result,*

$$\|\sqrt{n}\Omega_{jj}^{-1/2}\Big(\frac{1}{n}\sum_{i=1}^{n}g_ig_i^T\Big)^{-1/2}\Big(\frac{1}{n}\sum_{i=1}^{n}V_i\Big)\hat{B}^Te_j\|^2 \xrightarrow{\text{d}} \chi_K^2. \qquad (S3.6)$$

The proof of Theorem S3.1 is deferred to Section S5 and Section S6. In the next subsection, we prove Theorem 2.1 using Theorem S3.1.

## S3.2 Proof of Theorem 2.1

We restate Theorem 2.1 for convenience.

**Theorem 2.1.** *Let Assumptions 2.1 to 2.4 be fulfilled. Then for any $j \in [p]$ such that $H_0$ in (1.10) holds, and any minimizer $\hat{\mathsf{B}}$ of (1.5), we have*

$$\sqrt{\frac{n}{\Omega_{jj}}}\underbrace{\Big(\Big(\frac{1}{n}\sum_{i=1}^{n}\mathsf{g}_i\mathsf{g}_i^T\Big)^{1/2}\Big)^{\dagger}}_{\text{scalar}}\underbrace{\Big(\frac{1}{n}\sum_{i=1}^{n}\mathsf{V}_i\Big)}_{\mathbb{R}^{(K+1)\times(K+1)}}\underbrace{\hat{\mathsf{B}}^Te_j}_{\mathbb{R}^{K+1}} \xrightarrow{\text{d}} N\Big(\mathbf{0}, \ \underbrace{I_{K+1} - \frac{\mathbf{1}\mathbf{1}^T}{K+1}}_{\text{cov. } \mathbb{R}^{(K+1)\times(K+1)}}\Big), \qquad (2.1)$$

*where $\mathsf{g}_i = -\mathsf{y}_i + \hat{\mathsf{p}}_i$ as in (1.7) and $\mathsf{V}_i = \mathsf{H}_i - (\mathsf{H}_i \otimes x_i^T)[\sum_{l=1}^{n}\mathsf{H}_l \otimes (x_lx_l^T)]^{\dagger}(\mathsf{H}_i \otimes x_i)$.*

The proof of Theorem 2.1 is a consequence of Theorem S3.1. To begin with, we state the following useful lemma.

**Lemma S3.2.** *For $\mathsf{V}_i$ and $V_i$ defined in Theorems 2.1 and S3.1, we have $V_i = Q^T\mathsf{V}_iQ$.*

*Proof of Lemma S3.2.* Since $H_i = Q^T\mathsf{H}_iQ$, we have

$$V_i = H_i - (H_i \otimes x_i^T)\Big[\sum_{i=1}^{n}H_i \otimes (x_ix_i^T)\Big]^{-1}(H_i \otimes x_i)$$

$$= H_i - [(Q^T\mathsf{H}_iQ) \otimes x_i^T]\Big[\sum_{i=1}^{n}(Q^T\mathsf{H}_iQ) \otimes (x_ix_i^T)\Big]^{-1}[(Q^T\mathsf{H}_iQ) \otimes x_i]$$

$$= H_i - Q^T(\mathsf{H}_i \otimes x_i^T)(Q \otimes I_p)\Big[(Q^T \otimes I_p)[\sum_{i=1}^{n}\mathsf{H}_i \otimes (x_ix_i^T)](Q \otimes I_p)\Big]^{-1}(Q^T \otimes I_p)(\mathsf{H}_i \otimes x_i)Q$$

$$= Q^T\mathsf{H}_iQ - Q^T(\mathsf{H}_i \otimes x_i^T)\Big[\sum_{i=1}^{n}(\mathsf{H}_i \otimes x_ix_i^T)\Big]^{\dagger}(\mathsf{H}_i \otimes x_i)Q$$

$$= Q^T\mathsf{V}_iQ,$$

where the penultimate equality is proved as follows.

Let $A = Q \otimes I_p$ and $\mathsf{D} = \sum_{i=1}^{n}\mathsf{H}_i \otimes (x_ix_i^T)$ only in the remaining of this proof. It remains to prove

$$A[A^T\mathsf{D}A]^{-1}A^T = \mathsf{D}^{\dagger}. \qquad (S3.7)$$

Since $\mathsf{H}_i\mathbf{1} = 0$, we have $\mathsf{D}(\mathbf{1} \otimes I_p) = 0$. Since $Q^T\mathbf{1} = 0$ by definition of $Q$, we have $A^T(\mathbf{1} \otimes I_p) = 0$. If we write the eigen-decomposition of $\mathsf{D}$ as $\mathsf{D} = \sum_{i=1}^{pK}\lambda_iu_iu_i^T$, then $u_i^T(\mathbf{1} \otimes I_p) = 0$. Hence, with $v_i = A^Tu_i$,

$$A^T\mathsf{D}A = \sum_{i=1}^{pK}\lambda_iv_iv_i^T.$$

Since $v_i^T v_{i'} = u_i^T AA^T u_{i'} = u_i^T[(I_{K+1} - \frac{\mathbf{1}\mathbf{1}^T}{K+1}) \otimes I_p]u_{i'} = u_i^T u_{i'} = I(i = i')$, we have

$$A[A^T \mathsf{D} A]^{-1}A^T = A\Big(\sum_{i=1}^{pK} \lambda_i^{-1} v_i v_i^T\Big)A^T = \sum_{i=1}^{pK} \lambda_i^{-1} u_i u_i^T = \mathsf{D}^\dagger,$$

where the second equality uses $Av_i = AA^T u_i = u_i$. The proof of (S3.7) is complete. $\qquad\square$

Now we are ready to prove that Theorem 2.1 is a consequence of Theorem S3.1.

*Proof of Theorem 2.1.* By definition of $\mathsf{g}_i$ and $\mathsf{V}_i$, we have $\mathbf{1}^T\mathsf{g}_i = 0$ and $\mathbf{1}^T\mathsf{V}_i = \mathbf{0}^T$. Thus, we have $QQ^T\mathsf{g}_i = \mathsf{g}_i$ and $QQ^T\mathsf{V}_i = \mathsf{V}_i$. Therefore, we can rewrite the left-hand side of (2.1) (without $\sqrt{n}\Omega_{jj}^{-1/2}$) as

$$\begin{aligned}
&\Big(\Big(\frac{1}{n}\sum_{i=1}^n \mathsf{g}_i\mathsf{g}_i^T\Big)^{1/2}\Big)^\dagger\Big(\frac{1}{n}\sum_{i=1}^n \mathsf{V}_i\Big)\hat{\mathsf{B}}^T e_j \\
={}&\Big(\Big(\frac{1}{n}\sum_{i=1}^n QQ^T\mathsf{g}_i\mathsf{g}_i^T QQ^T\Big)^{1/2}\Big)^\dagger\Big(\frac{1}{n}\sum_{i=1}^n QQ^T\mathsf{V}_i QQ^T\Big)\hat{\mathsf{B}}^T e_j \\
={}&\Big(\Big(\frac{1}{n}\sum_{i=1}^n Qg_i g_i^T Q^T\Big)^{1/2}\Big)^\dagger\Big(\frac{1}{n}\sum_{i=1}^n QV_iQ^T\Big)\hat{\mathsf{B}}^T e_j \\
={}&Q\Big(\Big(\frac{1}{n}\sum_{i=1}^n g_i g_i^T\Big)^{1/2}\Big)^\dagger Q^T Q\Big(\frac{1}{n}\sum_{i=1}^n V_iQ^T\Big)\hat{\mathsf{B}}^T e_j \\
={}&Q\Big(\Big(\frac{1}{n}\sum_{i=1}^n g_i g_i^T\Big)^{1/2}\Big)^\dagger\Big(\frac{1}{n}\sum_{i=1}^n V_i\Big)\hat{B}^T e_j,
\end{aligned}$$

where the first equality uses $QQ^T\mathsf{g}_i = \mathsf{g}_i$ and $QQ^T\mathsf{V}_i = \mathsf{V}_i$, the second equality uses $g_i = Q^T\mathsf{g}_i$ and $V_i = Q^T\mathsf{V}_i Q$ from Lemma S3.2, the third equality follows from the same argument of (S3.7), and the last equality uses $Q^T Q = I_K$ and $\hat{B} = \hat{\mathsf{B}}Q$.

Therefore, Theorem S3.1 implies that the limiting covariance for the left-hand side of (2.1) is $QQ^T = I_K - \frac{\mathbf{1}\mathbf{1}^T}{K+1}$. This completes the proof. $\qquad\square$

# S4 Proof of Theorem 2.2

We restate Theorem 2.2 for convenience.

**Theorem 2.2.** *Define the matrix $R = (I_K, \mathbf{0}_K)^T \in \mathbb{R}^{(K+1)\times K}$ using block matrix notation. Let Assumptions 2.1 to 2.4 be fulfilled. For $\hat{A}$ in (1.9) and any $j \in [p]$ such that $H_0$ in (1.10) holds,*

$$\underbrace{\Big(I_K + \frac{\mathbf{1}_K\mathbf{1}_K^T}{\sqrt{K+1}+1}\Big)R^T}_{\text{matrix } \mathbb{R}^{K\times(K+1)}}\underbrace{\sqrt{\frac{n}{\Omega_{jj}}}}_{\text{scalar}}\underbrace{\Big(\Big(\frac{1}{n}\sum_{i=1}^n \mathsf{g}_i\mathsf{g}_i^T\Big)^{1/2}\Big)^\dagger}_{\text{matrix } \mathbb{R}^{(K+1)\times(K+1)}}\underbrace{\Big(\frac{1}{n}\sum_{i=1}^n \mathsf{V}_i R\Big)}_{\mathbb{R}^{(K+1)\times K}}\underbrace{\hat{A}^T e_j}_{\mathbb{R}^K}\xrightarrow{\text{d}} N(\mathbf{0}_K, I_K) \quad (2.2)$$

*where $\mathsf{g}_i$ is defined in (1.7) and $\mathsf{V}_i$ is defined in Theorem 2.1. Furthermore, for the same $j \in [p]$,*

$$\mathcal{T}_n^j(X, Y) \stackrel{\text{def}}{=} \frac{n}{\Omega_{jj}}\Big\|\Big(\Big(\frac{1}{n}\sum_{i=1}^n \mathsf{g}_i\mathsf{g}_i^T\Big)^{1/2}\Big)^\dagger\Big(\frac{1}{n}\sum_{i=1}^n \mathsf{V}_i\Big)R\hat{A}^T e_j\Big\|^2 \quad \text{satisfies} \quad \mathcal{T}_n^j(X, Y)\xrightarrow{\text{d}}\chi_K^2. \quad (2.3)$$

*Note that Equations (2.2) and (2.3) is stated using $\Omega_{jj} = e_j^T\Sigma^{-1}e_j$. When $\Sigma$ is unknown, the quantity $\Omega_{jj}$ in above results can be replaced by its consistent estimate $\hat{\Omega}_{jj}$ defined in (2.5), and the convergence in distribution results still hold.*

The proof is a direct consequence of Theorem 2.1.

*Proof of Theorem 2.2.* By definition of $\hat{A}$ in (1.9), we have $\hat{A} = \hat{\mathsf{B}}(I_K, -\mathbf{1}_K)^T$ and

$$\hat{A}\,(I_K, \mathbf{0}_K) = \hat{\mathsf{B}}(I_K, -\mathbf{1}_K)^T(I_K, \mathbf{0}_K) = \hat{\mathsf{B}}(I_{K+1} - e_{K+1}\mathbf{1}^T) = \hat{\mathsf{B}} - (\hat{\mathsf{B}}e_{K+1})\mathbf{1}^T,$$

which is of the form $\hat{\mathsf{B}} - b\mathbf{1}^T$ with $b = \hat{\mathsf{B}}e_{K+1}$. Therefore, $\hat{A}\,(I_K, \mathbf{0}_K)$ is also a solution of (1.5). Taking $\hat{\mathsf{B}}$ in Theorem 2.1 to be $\hat{A}\,(I_K, \mathbf{0}_K) = \hat{A}R^T$ gives the desired $\chi^2$ result (2.3) and

$$\sqrt{n}\Omega_{jj}^{-1/2}\left(\left(\frac{1}{n}\sum_{i=1}^{n}\mathsf{g}_i\mathsf{g}_i^T\right)^{1/2}\right)^{\dagger}\left(\frac{1}{n}\sum_{i=1}^{n}\mathsf{V}_i\right)R\hat{A}^Te_j \xrightarrow{\mathrm{d}} N\left(0, I_{K+1} - \frac{\mathbf{1}\mathbf{1}^T}{K+1}\right). \tag{S4.1}$$

Multiplying $(R^T(I_{K+1} - \frac{\mathbf{1}\mathbf{1}^T}{K+1})R)^{-1/2}R^T$ to the left of the above display gives the desired normality result (2.2) by observing $(R^T(I_{K+1} - \frac{\mathbf{1}\mathbf{1}^T}{K+1})R)^{-1/2} = (I_K + \frac{\mathbf{1}_K\mathbf{1}_K^T}{\sqrt{K+1}+1})$. This completes the proof. $\qquad\square$

## S5 Preliminary results for proving Theorem S3.1

### S5.1 Results for general loss functions

In this subsection, we will work under the following assumptions with a general convex loss function. Later in Section S6, we will apply the general results of this subsection to the multinomial logistic loss discussed in Section S3.1.

**Assumption S5.1.** Suppose we have data $(Y, X)$, where $Y \in \mathbb{R}^{n \times (K+1)}$ with rows $(\mathsf{y}_1, ..., \mathsf{y}_n)$, and $X \in \mathbb{R}^{n \times p}$ has i.i.d. rows $(x_1, ..., x_n)$ with $x_i \sim N(\mathbf{0}, \Sigma)$ and invertible $\Sigma$. The observations $(\mathsf{y}_i, x_i)_{i \in [n]}$ are i.i.d. and $\mathsf{y}_i$ has the form $\mathsf{y}_i = f(U_i, x_i^T B^*)$ for some deterministic function $f$, deterministic $B^* \in \mathbb{R}^{p \times K}$, and latent random variable $U_i$ independent of $x_i$. Assume $p/n \leq \delta^{-1} < 1$.

**Assumption S5.2.** Given data $(Y, X)$, consider twice continuously differentiable and strictly convex loss functions $(L_i)_{i \in [n]}$ with each $L_i : \mathbb{R}^K \to \mathbb{R}$ depending on $\mathsf{y}_i$ but not on $x_i$.

Provided that the following minimization problem admits a solution, define

$$\hat{B}(Y, X) = \arg\min_{B \in \mathbb{R}^{p \times K}} \sum_{i=1}^{n} L_i(B^T x_i).$$

Define for each $i \in [n]$,

$$g_i(Y, X) = \nabla L_i(\hat{B}(Y, X)^T x_i), \quad H_i(Y, X) = \nabla^2 L_i(\hat{B}(Y, X)^T x_i),$$

so that $g_i(Y, X) \in \mathbb{R}^K$ and $H_i(Y, X) \in \mathbb{R}^{K \times K}$. Define

$$G(Y, X) = \sum_{i=1}^{n} e_i g_i(Y, X)^T,$$

$$V(Y, X) = \sum_{i=1}^{n}\left(H_i(Y, X) - (H_i(Y, X) \otimes x_i^T)\left[\sum_{l=1}^{n} H_l(Y, X) \otimes (x_l x_l^T)\right]^{\dagger}(H_i(Y, X) \otimes x_i)\right),$$

so that $G(Y, X) \in \mathbb{R}^{n \times K}$ and $V(Y, X) \in \mathbb{R}^{K \times K}$. If the dependence on data $(Y, X)$ is clear from context, we will simply write $\hat{B}, g_i, H_i\ G$, and $V$.

**Theorem S5.1.** *Let Assumptions S5.1 and S5.2 be fulfilled. Let $c_*, m_*, m^*, K$ be positive constants independent of $n, p$. Let $U^* \subset \mathbb{R}^{p \times (K+1)} \times \mathbb{R}^{n \times p}$ be an open set satisfying*

    *(1) If $\{(Y, X) \in U^*\}$, then the minimizer $\hat{B}(Y, X)$ in Assumption S5.2 exists, $H_i \preceq I_K$ for each $i \in [n]$, $\frac{1}{n}\sum_{i=1}^{n} H_i(Y, X) \otimes (x_i x_i^T) \succeq c_*(I_K \otimes \Sigma)$ and $m_* I_K \preceq \frac{1}{n}G(Y, X)^T G(Y, X) \preceq m^* I_K$.*

    *(2) For any $\{(Y, X), (Y, \tilde{X})\} \subset U^*$, $\|G(Y, X) - G(Y, \tilde{X})\|_F \leq L\|(X - \tilde{X})\Sigma^{-1/2}\|_F$ holds for some positive constant $L$.*

*Then for any $j \in [p]$ such that $e_j^T B^* = \mathbf{0}_K^T$, there exists a random variable $\xi \in \mathbb{R}^K$ such that*

$$\mathbb{E}\left[I\{(Y, X) \in U^*\}\left\|\frac{(G^T G)^{-1/2} V \hat{B}^T e_j}{\sqrt{\Omega_{jj}}} - \xi\right\|^2\right] \leq \frac{C}{p-K},$$

*and $\mathbb{P}(\|\xi\|^2 > \chi_K^2(\alpha)) \leq \alpha$ for all $\alpha \in (0, 1)$, $C$ is a positive constant depending on $(c_*, m_*, m^*, K, L)$ only. If additionally $\mathbb{P}((Y, X) \in U^*) \to 1$, then $\xi$ in the previous display satisfies $\xi \xrightarrow{\text{d}} N(\mathbf{0}, I_K)$ and*

$$\frac{(G^T G)^{-1/2} V \hat{B}^T e_j}{\sqrt{\Omega_{jj}}} \xrightarrow{\text{d}} N(\mathbf{0}, I_K).$$

The proof of Theorem S5.1 is given in next subsection.

### S5.2   Proof of Theorem S5.1

In this subsection and next subsection, we will slightly abuse the notations $A^*$ and $\hat{A}$, which have different definitions than the definitions in the main text.

Let $\Sigma^{1/2} B^* = \sum_{k=1}^K s_k u_k v_k^T$ be the singular value decomposition of $\Sigma^{1/2} B^*$, where $u_1, ..., u_K$ are the left singular vectors and $v_1, ..., v_k$ the right singular vectors. If $\Sigma^{1/2} B^*$ is of rank strictly less than $K$, we allow some $s_k$ to be equal to 0 so that $\Sigma^{1/2} B^* = \sum_{k=1}^K s_k u_k v_k^T$ still holds with orthonormal $(u_1, ..., u_K)$ and orthonormal $(v_1, ..., v_K)$. We consider an orthogonal matrix $\tilde{P} \in \mathbb{R}^{p \times p}$ such that

$$\tilde{P}\tilde{P}^T = \tilde{P}^T \tilde{P} = I_p, \quad \tilde{P}\frac{\Sigma^{-1/2} e_j}{\|\Sigma^{-1/2} e_j\|} = e_1, \quad \tilde{P}u_k = e_{p-K+k}, \quad \forall k \in [K]. \tag{S5.1}$$

Since $e_j^T B^* = \mathbf{0}^T$ implies $e_j^T \Sigma^{-1/2} u_k = 0$, we can always find a matrix $\tilde{P}$ satisfying (S5.1). From now on we fix this matrix $\tilde{P}$ and consider the following change of variable,

$$Z = X \Sigma^{-1/2} \tilde{P}^T, \qquad A^* = \tilde{P} \Sigma^{1/2} B^*. \tag{S5.2}$$

It immediately follows that $Z$ has i.i.d. $N(0, 1)$ entries and the first $p - K$ rows of $A^*$ are all zeros. Since the response $\mathsf{y}_i$ has the expression $\mathsf{y}_i = f(U_i, x_i^T B^*)$, $Y$ is unchanged by the change of variable (S5.2) from $ZA^* = XB^*$. We now work on the multinomial logistic estimation with data $(Y, Z)$ and the underlying coefficient matrix $A^*$ in (S5.2). Parallel to the estimate $\hat{B}$ of $B^*$ in Assumption S5.2, we define the estimate of $A^*$ using data $(Y, Z)$ as

$$\hat{A}(Y, Z) = \arg\min_{A \in \mathbb{R}^{p \times K}} \sum_i L_i(A^T z_i),$$

where $z_i = Z^T e_i$ is the $i$-th row of $Z$. By construction, we have $\hat{A} = \tilde{P} \Sigma^{1/2} \hat{B}$, hence $Z\hat{A} = X\hat{B}$ and $e_1^T \hat{A} = e_j^T \hat{B} / \sqrt{\Omega_{jj}}$. Furthermore, the quantities depending on $(Y, X\hat{B})$ remain unchanged after the change of variable. In particular, the gradient and Hessian

$$\nabla L_i(\hat{B}^T x_i) = \nabla L_i(\hat{A}^T z_i), \qquad \nabla^2 L_i(\hat{B}^T x_i) = \nabla^2 L_i(\hat{A}^T z_i)$$

are unchanged. It follows that the matrix $G$ and $V$ are unchanged. Therefore, we have

$$\frac{e_j^T \hat{B} V (G^T G)^{-1/2}}{\sqrt{\Omega_{jj}}} = e_1^T \hat{A} V (G^T G)^{-1/2}.$$

In conclusion, with the change of variables (S5.2), we only need to prove Theorem S5.1 in the special case, where the design matrix $X$ i.i.d. $N(0, 1)$ entries and the response $Y$ is independent of the first $p - K$ columns of $X$. To this end, we introduce the following Theorem S5.2, and the proof of Theorem S5.1 is a consequence of Theorem S5.2 as it proves the desired result for $e_1^T \hat{A} V (G^T G)^{-1/2}$.

**Theorem S5.2.** *Let $c_*, m_*, m^*, K$ be constants independent of $n, p$. Let $Z \in \mathbb{R}^{n \times p}$ have i.i.d. rows $(z_1, ..., z_n)$ with $z_i \sim N(\mathbf{0}, I_p)$. Let $\mathsf{y}_1, ..., \mathsf{y}_n \in \mathbb{R}^{(K+1)}$ such that $(\mathsf{y}_1, ..., \mathsf{y}_n)$ is independent of the first $p - K$ columns of $Z$. Consider twice continuously differentiable and strictly convex loss*

*functions $(L_i)_{i=1,\ldots,n}$ with each $L_i : \mathbb{R}^K \to \mathbb{R}$ depending on $y_i$ but not on $z_i$ and define, provided that the minimizer admits a solution,*

$$\hat{A}(Y,Z) = \underset{A \in \mathbb{R}^{p \times K}}{\arg\min} \sum_{i=1}^{n} L_i(A^T z_i), \quad g_i(Y,Z) = \nabla L_i(\hat{A}(Y,Z)^T z_i), \quad H_i(Y,Z) = \nabla^2 L_i(\hat{A}(Y,Z)^T z_i),$$

*$G(Y,Z) = \sum_{i=1}^{n} e_i g_i(Y,Z)^T \in \mathbb{R}^{n \times K}$, and $V(Y,Z) = \sum_{i=1}^{n}\big(H_i - (H_i \otimes z_i^T)\big[\sum_{l=1}^{n} H_l \otimes (z_l z_l^T)\big]^\dagger (H_i \otimes z_i)\big) \in \mathbb{R}^{K \times K}$, where we dropped the dependence of $H_i$ on $(Y,Z)$ for simplicity.*

*Let $O \subset \mathbb{R}^{n \times (K+1)} \times \mathbb{R}^{n \times p}$ be an open set satisfying*

- *If $(Y,Z) \in O$, then the minimizer $\hat{A}(Y,Z)$ exists, $H_i \preceq I_K$ for each $i \in [n]$, $c_* I_{pK} \preceq \frac{1}{n}\sum_{i=1}^{n} H_i(Y,Z) \otimes (z_i z_i^T)$, and $m_* I_K \preceq \frac{1}{n}\sum_{i=1}^{n} G(Y,Z)^T G(Y,Z) \preceq m^* I_K$.*

- *With the notation $G(Y,Z) = \sum_{i=1}^{n} e_i g_i(Y,Z)^T$, we have if two $Z, \tilde{Z} \in \mathbb{R}^{n \times p}$ satisfy $\{(Y,Z),(Y,\tilde{Z})\} \subset O$ then $\|G(Y,Z) - G(Y,\tilde{Z})\| \leq L\|Z - \tilde{Z}\|$.*

*For $e_1 \in \mathbb{R}^p$ the first canonical basis vector, there exists a random variable $\xi \in \mathbb{R}^K$ such that*

$$\mathbb{E}\big[I\{(Y,Z) \in O\}\big\|(G^T G)^{-1/2} V \hat{A}^T e_1 - \xi\big\|^2\big] \leq \frac{C}{p-K},$$

*and $\mathbb{P}(\|\xi\|^2 > \chi_K^2(\alpha)) \leq \alpha$ for all $\alpha \in (0,1)$, $C$ is a positive constant depending on $(c_*, m_*, m^*, K, L)$ only. If additionally $\mathbb{P}((Y,Z) \in O) \to 1$, then $\xi$ in the previous display satisfies $\xi \xrightarrow{d} N(\mathbf{0}, I_K)$ and*

$$e_1^T \hat{A} V (G^T G)^{-1/2} \xrightarrow{d} N(\mathbf{0}, I_K).$$

The proof of Theorem S5.2 is presented in Section S5.3.

## S5.3   Proof of Theorem S5.2

We first present a few useful lemmas, whose proofs are given at the end of this subsection.

**Lemma S5.3** (Proof is given on page 24)**.** *Let $z \sim N(\mathbf{0}, \sigma^2 I_n)$ and $F : \mathbb{R}^n \to \mathbb{R}^{n \times K}$ be weakly differentiable with $\mathbb{E}\|F(z)\|_F^2 < \infty$. Let $\tilde{z}$ be an independent copy of $z$. Then*

$$\mathbb{E}\Big[\Big\|z^T F(z) - \sigma^2 \sum_{i=1}^{n} \frac{\partial e_i^T F(z)}{\partial z_i} - z^T F(\tilde{z})\Big\|^2\Big] \leq 3\sigma^4 \mathbb{E}\sum_{i=1}^{n}\Big\|\frac{\partial F(z)}{\partial z_i}\Big\|_F^2.$$

**Lemma S5.4** (Proof is given on page 25)**.** *If $G, \tilde{G} \in \mathbb{R}^{n \times K}$ satisfy $m_* I_K \preceq \frac{1}{n} G^T G \preceq m^* I_K$ and $m_* I_K \preceq \frac{1}{n}\tilde{G}^T \tilde{G} \preceq m^* I_K$ for some positive constants $m_*$ and $m^*$. Then*

$$\|(G^T G)^{-1/2} - (\tilde{G}^T \tilde{G})^{-1/2}\|_F \leq L_1 n^{-1}\|G - \tilde{G}\|_F,$$
$$\|G(G^T G)^{-1/2} - \tilde{G}(\tilde{G}^T \tilde{G})^{-1/2}\|_F \leq L_2 n^{-1/2}\|G - \tilde{G}\|_F,$$

*where $L_1, L_2$ are positive constants depending on $(K, m_*, m^*)$ only.*

**Lemma S5.5** (Proof is given on page 26)**.** *Let the assumptions in Theorem S5.2 be fulfilled. Let $Y \in \mathbb{R}^{n \times (K+1)}$ be fixed. If a minimizer $\hat{A}(Y,Z)$ exists at $Z$, then $Z \mapsto \hat{A}(Y,Z)$ exists and is differentiable in a neighborhood of $Z$ with derivative*

$$\frac{\partial \operatorname{vec}(\hat{A})}{\partial z_{ij}} = -M[g_i \otimes e_j + (H_i \hat{A}^T e_j \otimes z_i)],$$
$$\frac{\partial g_l}{\partial z_{ij}} = -(H_l \otimes z_l^T)M[g_i \otimes e_j + (H_i \hat{A}^T e_j \otimes z_i)] + I(l = i)H_l \hat{A}^T e_j,$$

*where $M = [\sum_{i=1}^{n} H_i \otimes (z_i z_i^T)]^{-1}$. It immediately follows that*

$$\frac{\partial g_i}{\partial z_{ij}} = [H_i - (H_i \otimes z_i^T)M(H_i \otimes z_i)]\hat{A}^T e_j - (H_i \otimes z_i^T)M(g_i \otimes e_j).$$

**Corollary S5.6** (Proof is given on page 26). *Under the same conditions of Lemma S5.5, for $G = \sum_{i=1}^{n} e_i g_i^T$, we have for each $i \in [n], j \in [p]$,*

$$\sum_{i=1}^{n} \frac{\partial e_i^T G (G^T G)^{-1/2}}{\partial z_{ij}}$$

$$= e_j^T \hat{A} V_i^T (G^T G)^{-1/2} + \sum_{i=1}^{n} \Big[ -(g_i^T \otimes e_j^T) M (H_i \otimes z_i)(G^T G)^{-1/2} + e_i^T G \frac{\partial (G^T G)^{-1/2}}{\partial z_{ij}} \Big].$$

Now we are ready to prove Theorem S5.2.

*Proof of Theorem S5.2.* Let $h : O \to \mathbb{R}^{n \times K}$ be $h(Y, Z) = G(Y, Z)(G(Y, Z)^T G(Y, Z))^{-1/2}$. In most of this proof, we will omit the dependence $(Y, Z)$ on $h, \hat{A}, g_i, H_i, G, V$ to lighten notation. By Lemma S5.4, we know this $h$ is $LL_2 n^{-1/2}$-Lipschitz in the sense that $\|h(Y, Z) - h(Y, \tilde{Z})\|_F \leq LL_2 n^{-1/2} \|Z - \tilde{Z}\|_F$ for all $\{(Y, Z), (Y, \tilde{Z})\} \subset O$. By Kirszbraun theorem, there exists a function $H : \mathbb{R}^{n \times (K+1)} \times \mathbb{R}^{n \times p} \to \mathbb{R}^{n \times K}$ (an extension of $h$ from $O$ to $\mathbb{R}^{n \times (K+1)} \times \mathbb{R}^{n \times p}$) such that $H(Y, Z) = h(Y, Z)$ for all $(Y, Z) \in O$, $\|H(Y, Z)\|_{op} \leq 1$ and $\|H(Y, Z) - H(Y, \tilde{Z})\|_F \leq LL_2 n^{-1/2} \|Z - \tilde{Z}\|_F$ for all $\{(Y, Z), (Y, \tilde{Z})\} \subset \mathbb{R}^{n \times (K+1)} \times \mathbb{R}^{n \times p}$.

For each $j \in [p]$, let $\mathsf{z}_j = Z e_j$ be the $j$-th column of $Z$ to distinguish it from the notation $z_i$, which means the $i$-th row of $Z$. Let $\check{z} \sim N(\mathbf{0}, I_n)$ be an independent copy of each columns of $Z$, and $\check{Z}^j = Z(I_p - e_j e_j^T) + \check{z} e_j^T$. That is, $\check{Z}^j$ replaces the $j$-th column of $Z$ by $\check{z}$. By definition, $\mathsf{z}_1 \perp \check{z}$ and $\mathsf{z}_1 \perp \check{Z}^1$.

Let $\xi = -[H(Y, \check{Z}^1)]^T \mathsf{z}_1 \in \mathbb{R}^K$, then $\|\xi\|^2 \leq \|\mathsf{z}_1\|$ since $\|H(Y, \check{Z}^1)\|_{op} \leq 1$. It follows that

$$\mathbb{P}(\|\xi\|^2 > \chi_K^2(\alpha)) \leq \mathbb{P}(\|\mathsf{z}_1\|^2 > \chi_K^2(\alpha)) = \alpha.$$

Note that the first $p - K$ columns of $Z$ are exchangeable, because they are i.i.d. and independent of the response $Y$, we have for each $\ell \in [p - K]$,

$$\mathbb{E}\Big[ I\{(Y, Z) \in O\} \big\| (G^T G)^{-1/2} V \hat{A}^T e_1 - \xi \big\|^2 \Big]$$
$$= \mathbb{E}\Big[ I\{(Y, Z) \in O\} \| e_1^T \hat{A} V (G^T G)^{-1/2} + \mathsf{z}_1^T H(Y, \check{Z}^1) \|^2 \Big]$$
$$= \mathbb{E}\Big[ I\{(Y, Z) \in O\} \| e_\ell^T \hat{A} V (G^T G)^{-1/2} + \mathsf{z}_\ell^T H(Y, \check{Z}^\ell) \|^2 \Big],$$

where the last line holds for any $\ell \in [p - K]$ because $(\mathsf{z}_1, e_1^T \hat{A}, \check{Z}) \stackrel{d}{=} (\mathsf{z}_\ell, e_\ell^T \hat{A}, \check{Z}^\ell)$. Therefore,

$$\mathbb{E}\Big[ I\{(Y, Z) \in O\} \big\| e_1^T \hat{A} V (G^T G)^{-1/2} + \mathsf{z}_1^T H(Y, \check{Z}^1) \big\|^2 \Big]$$
$$= \frac{1}{p - K} \sum_{\ell=1}^{p-K} \mathbb{E}\Big[ I\{(Y, Z) \in O\} \big\| e_\ell^T \hat{A} V (G^T G)^{-1/2} + \mathsf{z}_\ell^T H(Y, \check{Z}^\ell) \big\|^2 \Big]$$
$$= \frac{1}{p - K} \sum_{\ell=1}^{p-K} \mathbb{E}\Big[ I\{(Y, Z) \in O\} \big\| \sum_i \frac{\partial e_i^T G (G^T G)^{-1/2}}{\partial z_{i\ell}} + \mathsf{z}_\ell^T H(Y, \check{Z}^\ell) - \mathrm{Rem}_\ell \big\|^2 \Big],$$

where $\mathrm{Rem}_\ell = \sum_{i=1}^{n} \big[ -(g_i^T \otimes e_\ell^T) M (H_i \otimes z_i)(G^T G)^{-1/2} + e_i^T G \frac{\partial (G^T G)^{-1/2}}{\partial z_{i\ell}} \big]$ from Corollary S5.6, and $M = [\sum_{i=1}^{n} (H_i \otimes z_i z_i^T)]^{-1}$ from Lemma S5.5. Using $(a + b)^2 \leq 2a^2 + 2b^2$, the above display can be bounded by sum of two terms, denoted by $(RHS)_1$ and $(RHS)_2$.

For the first term,

$$(RHS)_1 = \frac{2}{p - K} \sum_{\ell=1}^{p-K} \mathbb{E}\Big[ I\{(Y, Z) \in O\} \big\| \sum_i \frac{\partial e_i^T h(Y, Z)}{\partial z_{i\ell}} + \mathsf{z}_\ell^T H(Y, \check{Z}^\ell) \big\|^2 \Big].$$

Let $F(z_\ell) = H(Y, Z(I - e_\ell e_\ell^T) + z_\ell e_\ell^T) = H(Y, Z)$, then $F(\check{z}) = H(Y, \check{Z}^\ell)$. Apply Lemma S5.3 to $F(z_\ell)$ conditionally on $Z(I - e_\ell e_\ell^T)$, we obtain

$$\mathbb{E}\Big[I\{(Y, Z) \in O\}\Big\|\sum_i \frac{\partial e_i^T h(Y, Z)}{\partial z_{i\ell}} + z_\ell^T F(\check{z})\Big\|^2\Big]$$

$$= \mathbb{E}\Big[I\{(Y, Z) \in O\}\Big\|z_\ell^T F(z_\ell) - \sum_i \frac{\partial e_i^T F(z_\ell)}{\partial z_{i\ell}} - z_\ell^T F(\check{z})\Big\|^2\Big]$$

$$\leq \mathbb{E}\Big[\Big\|z_\ell^T F(z_\ell) - \sum_i \frac{\partial e_i^T F(z_\ell)}{\partial z_{i\ell}} - z_\ell^T F(\check{z})\Big\|^2\Big]$$

$$\leq 3\sum_i \mathbb{E}\Big\|\frac{\partial F(z_\ell)}{\partial z_{i\ell}}\Big\|_F^2$$

$$= 3\sum_i \mathbb{E}\Big\|\frac{\partial H(Y, Z)}{\partial z_{i\ell}}\Big\|_F^2,$$

where the first equality uses $z_\ell^T h(Y, Z) = 0$ from the KKT conditions $Z^T G = 0$ and $h(Y, Z) = G(G^T G)^{-1/2}$. It follows that

$$(RHS)_1 \leq \frac{6}{p - K}\mathbb{E}\Big[\sum_{\ell=1}^p \sum_{i=1}^n \Big\|\frac{\partial H(Y, Z)}{\partial z_{i\ell}}\Big\|_F^2\Big].$$

Note that the integrand in the last display is actually the squared Frobenius norm of the Jacobian of the mapping from $\mathbb{R}^{n \times p}$ to $\mathbb{R}^{n \times K}$: $Z \mapsto H(Y, Z)$. This Jacobian is a matrix with $nK$ rows and $np$ columns, has rank at most $nK$ and operator norm at most $LL_2 n^{-1/2}$ because $Z \mapsto H(Y, Z)$ is $LL_2 n^{-1/2}$-Lipschitz from Lemma S5.4. Using $\|A\|_F^2 \leq \text{rank}(A)\|A\|_{op}^2$, we obtain

$$(RHS)_1 \leq 6K(LL_2)^2/(p - K).$$

For the second term $(RHS)_2 = \frac{2}{p - K}\sum_{\ell=1}^{p-K} \mathbb{E}\big[I\{(Y, Z) \in O\}\|\text{Rem}_\ell\|^2\big]$. By definition of $\text{Rem}_\ell$ and $(a + b)^2 \leq 2a^2 + 2b^2$, we obtain

$$(RHS)_2 \leq \frac{4}{p - K}\sum_{\ell=1}^{p-K} \mathbb{E}\Big[I\{(Y, Z) \in O\}\Big\|\sum_{i=1}^n (g_i^T \otimes e_\ell^T)M(H_i \otimes z_i)(G^T G)^{-1/2}\Big\|^2\Big] \quad \text{(S5.3)}$$

$$+ \frac{4}{p - K}\sum_{\ell=1}^{p-K} \mathbb{E}\Big[I\{(Y, Z) \in O\}\Big\|\sum_{i=1}^n e_i^T G\frac{\partial(G^T G)^{-1/2}}{\partial z_{i\ell}}\Big\|^2\Big]. \quad \text{(S5.4)}$$

We next bound (S5.3) and (S5.4) one by one. For (S5.3), we focus on the norm without $(G^T G)^{-1/2}$ which is $\|\sum_{i=1}^n (g_i^T \otimes e_\ell^T)M(H_i \otimes z_i)\|$. With $\|a\| = \max_{u:\|u\|=1} a^T u$ in mind, let us multiply to the right by a unit vector $u \in \mathbb{R}^K$ and instead bound

$$\sum_{i=1}^n (g_i^T \otimes e_\ell^T)M(H_i \otimes z_i)u = \text{Tr}\Big[(I_K \otimes e_\ell^T)M\sum_i (H_i u \otimes z_i)g_i^T\Big] \leq K\|(I_K \otimes e_\ell^T)M\|_{op}\|\sum_i (H_i u \otimes z_i)g_i^T\|_{op}$$

because the rank of the matrix inside the trace is at most $K$ and $\text{Tr}[\cdot] \leq K\|\cdot\|_{op}$ holds. Then

$$\|\sum_i (H_i u \otimes z_i)g_i^T\|_{op} = \|(I_K \otimes Z^T)\sum_i (H_i u \otimes e_i)e_i^T G\|_{op} \leq \|Z\|_{op}\|\sum_i (H_i u \otimes e_i e_i^T)\|_{op}\|G\|_{op}.$$

Next, $\|\sum_i (H_i u \otimes e_i e_i^T)\|_{op} = \|\sum_i (H_i \otimes e_i e_i^T)(u \otimes I_n)\|_{op} \leq 1$ because $H_i \preceq I_K$ and $\|u\| = 1$. In summary, the norm in (S5.3) is bounded from above by

$$K\|(G^T G)^{-1/2}\|_{op}\|M\|_{op}\|Z\|_{op}\|G\|_{op}.$$

To bound (S5.3), since in the event $(Y, Z) \in O$, $m_* I_K \preceq \frac{1}{n}G^T G \preceq m^* I_K$ and $\frac{1}{n}\sum_{i=1}^n (H_i \otimes z_i z_i^T) \succeq c_* I_{pK}$, we have $\|G\|_{op} \leq \sqrt{m^* n}$, hence $\|M\|_{op} \leq c_*^{-1}$. Thus, the above display can be bounded by

$$K(m_* n)^{-1/2}c_*^{-1}\|Z\|_{op}\sqrt{m^* n} = (m^*/m_*)^{1/2}c_*^{-1}K\|Z\|_{op}.$$

Since $Z \in \mathbb{R}^{n \times p}$ has i.i.d. $N(0,1)$ entries, [Davidson and Szarek, 2001, Theorem II.13] implies that $\mathbb{E}\|Z\|_{op} \le \sqrt{n} + \sqrt{p} \le 2\sqrt{n}$. Therefore,

$$(S5.3) \le C(c_*, K, L)n^{-1}.$$

Now we bound (S5.4). Since

$$\sum_{\ell=1}^{p-K} \Big\| \sum_{i=1}^{n} e_i^T G \frac{\partial (G^T G)^{-1/2}}{\partial z_{i\ell}} \Big\|^2$$

$$\le \sum_{j=1}^{p} \Big\| \sum_{i=1}^{n} e_i^T G \frac{\partial (G^T G)^{-1/2}}{\partial z_{ij}} \Big\|^2$$

$$= \sum_{j=1}^{p} \sum_{k'=1}^{K} \Big( \sum_{i=1}^{n} \sum_{k=1}^{K} e_i^T G e_k e_k^T \frac{\partial (G^T G)^{-1/2}}{\partial z_{ij}} e_{k'} \Big)^2$$

$$\le \sum_{j=1}^{p} \sum_{k'=1}^{K} \Big[ \sum_{i=1}^{n} \sum_{k=1}^{K} (e_i^T G e_k)^2 \sum_{i=1}^{n} \sum_{k=1}^{K} \Big( e_k^T \frac{\partial (G^T G)^{-1/2}}{\partial z_{ij}} e_{k'} \Big)^2 \Big]$$

$$= \|G\|_F^2 \sum_{i=1}^{n} \sum_{j=1}^{p} \Big\| \frac{\partial (G^T G)^{-1/2}}{\partial z_{ij}} \Big\|^2.$$

Using $\|G\|_F^2 \le nK$, and the mapping $Z \mapsto (G^T G)^{-1/2}$ is $LL_1 n^{-1}$- Lipschitz on $O$ using Lemma S5.4, we conclude that

$$(S5.4) \le 4K^3 LL_1/(p-K).$$

Combining the above bounds on $(RHS)_1$ and $(RHS)_2$, we have

$$\mathbb{E}\Big[ I\{(Y,Z) \in O\} \|(G^T G)^{-1/2} V \hat{A}^T e_1 - \xi\|^2 \Big] \le \frac{C(c_*, K, m_*, m^*, L, L_1, L_2)}{p-K}, \qquad (S5.5)$$

where the constant depends on $(c_*, K, m_*, m^*, L)$ only because $L_1$ and $L_2$ are constants depending on $(K, m_*, m^*)$ only.

If additionally $\mathbb{P}((Y,Z) \in O) \to 1$, we have $\mathbb{P}((Y, \check{Z}^1) \in O) \to 1$ using $(Y,Z) \stackrel{d}{=} (Y, \check{Z}^1)$. Therefore,

$$\xi = -[h(Y, \check{Z}^1)^T \mathsf{z}_1] I((Y, \check{Z}^1) \in O) - [H(Y, \check{Z}^1)^T \mathsf{z}_1] I((Y, \check{Z}^1) \notin O) \stackrel{d}{\longrightarrow} N(\mathbf{0}, I_K). \qquad (S5.6)$$

By (S5.5), we know $(G^T G)^{-1/2} V \hat{A}^T e_1 - \xi \stackrel{d}{\longrightarrow} 0$ when $\mathbb{P}((Y,Z) \in O) \to 1$. Hence, we conclude

$$(G^T G)^{-1/2} V \hat{A}^T e_1 \stackrel{d}{\longrightarrow} N(\mathbf{0}, I_K) \quad \text{and} \quad \|(G^T G)^{-1/2} V \hat{A}^T e_1\|^2 \stackrel{d}{\longrightarrow} \chi_K^2.$$

$$\square$$

We next prove Lemmas S5.3 to S5.5 and corollary S5.6.

*Proof of Lemma S5.3.* Let $z_0 = (z^T, \tilde{z}^T)^T \in \mathbb{R}^{2n}$, then $z_0 \sim N(\mathbf{0}, \sigma^2 I_{2n})$. For each $k \in [K]$, let $f^{(k)} : \mathbb{R}^{2n} \to \mathbb{R}^{2n}$ be

$$f^{(k)}(z_0) = \begin{pmatrix} [F(z) - F(\tilde{z})]e_k \\ 0_n \end{pmatrix},$$

so that $z_0^T f^{(k)}(z_0) = z^T [F(z) - F(\tilde{z})]e_k$, and $\operatorname{div} f^{(k)}(z_0) = \sum_{i=1}^{n} \frac{\partial e_i^T F(z)e_k}{\partial z_i}$. Applying the second order Stein formula [Bellec and Zhang, 2021] (see also [Tan et al., 2022, Lemma F.1] for a

collection of such formulas) to $f^{(k)}$ gives, with Jac denoting the Jacobian,

$$
\mathbb{E}\Big[\Big(z^T F(z)e_k - \sigma^2 \sum_{i=1}^{n} \frac{\partial e_i^T F(z)e_k}{\partial z_i} - z^T F(\tilde{z})e_k\Big)^2\Big]
$$

$$
= \mathbb{E}\Big[\Big(z_0^T f^{(k)}(z_0) - \sigma^2 \operatorname{div} f^{(k)}(z_0)\Big)^2\Big]
$$

$$
= \sigma^2 \mathbb{E}\|f^{(k)}(z_0)\|^2 + \sigma^4 \mathbb{E}\operatorname{Tr}[(\operatorname{Jac} f^{(k)}(z_0))^2]
$$

$$
= \sigma^2 \mathbb{E}\|[F(z) - F(\tilde{z})]e_k\|^2 + \sigma^4 \mathbb{E}\operatorname{Tr}\Big[\Big(\begin{matrix} \operatorname{Jac}[F(z)e_k] & -\operatorname{Jac}[F(\tilde{z})e_k] \\ 0_{n\times n} & 0_{n\times n} \end{matrix}\Big)^2\Big]
$$

$$
= 2\sigma^2 \mathbb{E}\|[F(z) - \mathbb{E}F(z)]e_k\|^2 + \sigma^4 \mathbb{E}\operatorname{Tr}((\operatorname{Jac}[F(z)e_k])^2)
$$

$$
\leq 3\sigma^4 \mathbb{E}\|\operatorname{Jac}[F(z)e_k]\|_F^2,
$$

where the last inequality uses the Gaussian Poincaré inequality, and the Cauchy-Schwarz inequality $\operatorname{Tr}(A^2) \leq \|A\|_F^2$. Summing over $k \in [K]$ gives the desired inequality. $\qquad\square$

*Proof of Lemma S5.4.* We first prove $G \mapsto G^T G$ is Lipschitz by noting

$$
\|G^T G - \tilde{G}^T \tilde{G}\|_{op}
$$

$$
= \|(G - \tilde{G})^T G + \tilde{G}^T(G - \tilde{G})\|_{op}
$$

$$
\leq \|G - \tilde{G}\|_{op}(\|G\|_{op} + \|\tilde{G}\|_{op})
$$

$$
\leq 2\sqrt{m^* n}\|G - \tilde{G}\|_{op}.
$$

Then we show $G^T G \mapsto (G^T G)^{-1}$ is Lipschitz. Let $A = G^T G$ and $\tilde{A} = \tilde{G}^T \tilde{G}$, we have

$$
\|A^{-1} - \tilde{A}^{-1}\|_{op}
$$

$$
= \|A^{-1}(\tilde{A} - A)\tilde{A}^{-1}\|_{op}
$$

$$
\leq \|A - \tilde{A}\|_{op}\|A^{-1}\|_{op}\|\tilde{A}^{-1}\|_{op}
$$

$$
\leq (m_* n)^{-2}\|A - \tilde{A}\|_{op}.
$$

We next prove $(G^T G)^{-1} \mapsto (G^T G)^{-1/2}$ is Lipschitz. Let $S = (G^T G)^{-1}$, $S' = (\tilde{G}^T \tilde{G})^{-1}$, and if $u$ with $\|u\| = 1$ is the eigenvector of $\sqrt{S} - \sqrt{\tilde{S}}$ with eigenvalue $d$, then

$$
u^T(S - \tilde{S})u = u^T(\sqrt{S} - \sqrt{\tilde{S}})\sqrt{S}u + u^T\sqrt{\tilde{S}}(\sqrt{S} - \sqrt{\tilde{S}})u
$$

$$
= du^T\sqrt{S}u + du^T\sqrt{\tilde{S}}u
$$

$$
= du^T(\sqrt{S} + \sqrt{\tilde{S}})u.
$$

As $d$ can be chosen as $\pm\|\sqrt{S} - \sqrt{\tilde{S}}\|_{op}$ (this argument is a special case of the Hemmen-Ando inequality [van Hemmen and Ando, 1980]), this implies

$$
\|\sqrt{S} - \sqrt{\tilde{S}}\|_{op} = \frac{|u^T(S - \tilde{S})u|}{u^T(\sqrt{S} + \sqrt{\tilde{S}})u} \leq \frac{\|S - \tilde{S}\|_{op}}{\lambda_{\min}(\sqrt{S} + \sqrt{\tilde{S}})} \leq \frac{\|S - \tilde{S}\|_{op}}{2/\sqrt{m^* n}}.
$$

Combining the above Lipschitz results, we have

$$
\|(G^T G)^{-1/2} - (\tilde{G}^T \tilde{G})^{-1/2}\|_{op} \leq (m^* n)^{1/2}(m_* n)^{-2}(m^* n)^{1/2}\|G - \tilde{G}\|_{op} = \frac{m^*}{m_*^2}n^{-1}\|G - \tilde{G}\|_{op}.
$$

It immediately follows that

$$
\|(G^T G)^{-1/2} - (\tilde{G}^T \tilde{G})^{-1/2}\|_F \leq \sqrt{K}\frac{m^*}{m_*^2}n^{-1}\|G - \tilde{G}\|_F.
$$

That is, the mapping $G \mapsto (G^T G)^{-1/2}$ is $L_1 n^{-1}$-Lipschitz, where $L_1 = \sqrt{K}m_*^{-2}m^*$.

For the second statement, the result follows by

$$\|G(G^T G)^{-1/2} - \tilde{G}(\tilde{G}^T \tilde{G})^{-1/2}\|_{op}$$
$$\leq \|G - \tilde{G}\|_{op}\|(G^T G)^{-1/2}\|_{op} + \|\tilde{G}\|_{op}\|(G^T G)^{-1/2} - (\tilde{G}^T \tilde{G})^{-1/2}\|_{op}$$
$$\leq \|G - \tilde{G}\|_{op}(m_* n)^{-1/2} + (m^* n)^{1/2} L_1 n^{-1} \|G - \tilde{G}\|_{op}$$

Hence,

$$\|G(G^T G)^{-1/2} - \tilde{G}(\tilde{G}^T \tilde{G})^{-1/2}\|_F$$
$$\leq \sqrt{K}\big(m_*^{-1/2} + (m^*)^{1/2} L_1\big) n^{-1/2} \|G - \tilde{G}\|_F,$$

where $L_2 = \sqrt{K}\big(m_*^{-1/2} + (m^*)^{1/2} L_1\big)$. $\qquad\square$

*Proof of Lemma S5.5.* Recall the KKT conditions $\sum_{l=1}^{n} z_l g_l^T = \mathbf{0}_{p \times K}$. We look for the derivative with respect to $z_{ij}$. Denoting derivatives with a dot, we find by the chain rule and product rule

$$\dot{z}_l = \frac{\partial z_l}{\partial z_{ij}} = I(l = i) e_j,$$

$$\dot{g}_l = \frac{\partial g_l}{\partial z_{ij}} = \frac{\partial g_l}{\partial \hat{A}^\top z_l} \frac{\partial \hat{A}^\top z_l}{\partial z_{ij}} = H_l[\dot{A}^\top z_l + I(l = i)\hat{A}^\top e_j].$$

Thus, differentiating the KKT conditions w.r.t. $z_{ij}$ by the product rule gives

$$\sum_{l=1}^{n}\big[I(l=i)e_j g_l^T + x_l\big(\dot{A}^\top z_l + I(l=i)\hat{A}^\top e_j\big)^T H_l\big] = 0.$$

That is,

$$e_j g_i^T + \sum_{l=1}^{n} z_l z_l^T \dot{A} H_l + z_i e_j^T \hat{A} H_i = 0.$$

We then move the term involving $\dot{A}$ to one side, and vectorize both sides,

$$g_i \otimes e_j + (H_i \hat{A}^T e_j \otimes z_i) = -\sum_{l=1}^{n}(H_l \otimes z_l z_l^T) \operatorname{vec}(\dot{A}).$$

With $M = [\sum_{l=1}^{n}(H_l \otimes z_l z_l^T)]^{-1}$, we obtain

$$\operatorname{vec}(\dot{A}) = -M[g_i \otimes e_j + (H_i \hat{A}^T e_j \otimes z_i)].$$

Hence, using $\operatorname{vec}(H_l \dot{A}^\top z_l) = \operatorname{vec}(z_l^T \dot{A} H_l) = (H_l \otimes z_l^T) \operatorname{vec}(\dot{A})$ gives

$$\dot{g}_l = (H_l \otimes z_l^T) \operatorname{vec}(\dot{A}) + I(l=i) H_l \hat{A}^T e_j$$
$$= -(H_l \otimes z_l^T) M[g_i \otimes e_j + (H_i \hat{A}^T e_j \otimes z_i)] + I(l=i) H_l \hat{A}^T e_j.$$

Thus,

$$\dot{g}_i = -(H_i \otimes z_i^T) M[g_i \otimes e_j + (H_i \hat{A}^T e_j \otimes z_i)] + H_i \hat{A}^T e_j$$
$$= -(H_i \otimes z_i^T) M(H_i \hat{A}^T e_j \otimes z_i) + H_i \hat{A}^T e_j - (H_i \otimes z_i^T) M(g_i \otimes e_j)$$
$$= [H_i - (H_i \otimes z_i^T) M(H_i \otimes z_i)] \hat{A}^T e_j - (H_i \otimes z_i^T) M(g_i \otimes e_j)$$
$$= V_i \hat{A}^T e_j - (H_i \otimes z_i^T) M(g_i \otimes e_j),$$

where $V_i = [H_i - (H_i \otimes z_i^T) M(H_i \otimes z_i)]$. $\qquad\square$

*Proof of Corollary S5.6.* For each $i \in [n], j \in [p]$, we have by the product rule

$$\frac{\partial e_i^T G(G^T G)^{-1/2}}{\partial z_{ij}}$$

$$= \frac{\partial g_i^T}{\partial z_{ij}}(G^T G)^{-1/2} + e_i^T G \frac{\partial (G^T G)^{-1/2}}{\partial z_{ij}}$$

$$= [V_i \hat{A}^T e_j - (H_i \otimes z_i^T) M (g_i \otimes e_j)]^T (G^T G)^{-1/2} + e_i^T G \frac{\partial (G^T G)^{-1/2}}{\partial z_{ij}}$$

$$= e_j^T \hat{A} V_i (G^T G)^{-1/2} + \left[ -(g_i^T \otimes e_j^T) M (H_i \otimes z_i)(G^T G)^{-1/2} + e_i^T G \frac{\partial (G^T G)^{-1/2}}{\partial z_{ij}} \right].$$

With $V = \sum_{i=1}^n V_i$, we further have

$$\sum_{i=1}^n \frac{\partial e_i^T G(G^T G)^{-1/2}}{\partial z_{ij}}$$

$$= e_j^T \hat{A} V_i^T (G^T G)^{-1/2} + \sum_{i=1}^n \left[ -(g_i^T \otimes e_j^T) M (H_i \otimes z_i)(G^T G)^{-1/2} + e_i^T G \frac{\partial (G^T G)^{-1/2}}{\partial z_{ij}} \right].$$

$\square$

# S6 Proof of Theorem S3.1

Recall that Theorem S5.1 holds for general loss function $L_i : \mathbb{R}^K \to \mathbb{R}$ provided that conditions (1) and (2) in Theorem S5.1 hold. In this section, we consider the multinomial logistic loss function $L_i$ defined in Section S3.1. To be specific,

$$L_i(u) = - \sum_{k=1}^{K+1} \mathsf{y}_{ik} e_k^T Q u + \log \sum_{k'=1}^{K+1} \exp(e_{k'}^T Q u), \qquad \forall u \in \mathbb{R}^K. \tag{S6.1}$$

In order to apply Theorem S5.1, we need to verify that, when $L_i$ in (S6.1) is used, the two conditions (1) and (2) in Theorem S5.1 hold. To this end, we present a few lemmas in the following two subsections, which will be useful for asserting the conditions (1) and (2) when we apply Theorem S5.1 to prove Theorem S3.1.

## S6.1 Control of the singular values of the gradients and Hessians

Before stating the lemmas that assert the conditions in Theorem S5.1, define

$$U = \left\{ (Y, X) \in \mathbb{R}^{n \times (K+1)} \times \mathbb{R}^{n \times p} : \hat{\mathsf{B}} \text{ exists}, \| X\hat{\mathsf{B}}(I_{K+1} - \frac{\mathbf{1}\mathbf{1}^T}{K+1}) \|_F^2 < n\tau \right\},$$

$$U_y = \left\{ Y \in \mathbb{R}^{n \times (K+1)} : \sum_{i=1}^n I(\mathsf{y}_{ik} = 1) \geq \gamma n \text{ for all } k \in [K+1] \right\}.$$

**Lemma S6.1** (deterministic result on gradient)**.** *Let $L_i$ be defined as in* (S6.1)*. Assume that either Assumption 2.2 or Assumption S1.1 holds. If $Y \in U_y$, for any $M \in \mathbb{R}^{n \times K}$ such that $\|MQ^T\|_F^2 \leq n\tau$, we have*

$$m_* I_K \preceq n^{-1} \sum_{i=1}^n \nabla L_i(M^T e_i) \nabla L_i(M^T e_i)^T \preceq K I_K,$$

*where $m_*$ is a positive constant depending on $(K, \gamma, \tau)$ only.*

*Proof of Lemma S6.1.* Without loss of generality, let's assume that $\gamma n$ is an integer. Otherwise, we can replace it with the greatest integer less than or equal to $\gamma n$, denoted as $\lfloor \gamma n \rfloor$.

If $Y \in U_y$, there exists at least $\gamma n$ many disjoint index sets $\{S_1, \ldots, S_{\gamma n}\}$ such that the following hold for each $l \in [\gamma n]$,

$$(i) \ S_l \subset [n]; (ii) \ |S_l| = K+1; (iii) \ \sum_{i \in S_l} \mathsf{y}_{ik} = 1, \quad \forall k \in [K+1].$$

Since $S_l$ are disjoint and $\cup_{l=1}^{\gamma n} S_l \subset [n]$, we have

$$\sum_{l=1}^{\gamma n} \sum_{i \in S_l} \|QM^T e_i\|^2 \le \sum_{i=1}^{n} \|QM^T e_i\|^2 = \|QM^T\|_F^2 < n\tau.$$

It follows that at most $\alpha n$ many of $l \in \{1, 2, ..., \gamma n\}$ s.t. $\sum_{i \in S_l} \|QM^T e_i\|^2 > \tau/\alpha$, otherwise the previous display can not hold. In other words, there exists a subset $L^* \subset \{1, 2, ..., \gamma n\}$ with $|L^*| \ge (\gamma - \alpha)n$ s.t. $\sum_{i \in S_l} \|QM^T e_i\|^2 \le \tau/\alpha$ for all $l \in L^*$. Define the index set $I = \cup_{l \in L^*} S_l$, then $|I| \ge (K+1)n(\gamma - \alpha)$, and $\|QM^T e_i\|_\infty \le \sqrt{\tau/\alpha}$ for all $i \in I$. Let us take $\alpha = \gamma/2$, then $|L^*| \ge \frac{\gamma}{2}n$ and $|I| \ge \gamma(K+1)n/2$. Recall that $L_i(u) = \mathsf{L}_i(Qu)$, we have $\nabla L_i(u) = Q^T \nabla \mathsf{L}_i(Qu)$. Thus,

$$\nabla L_i(M^T e_i) = Q^T \nabla \mathsf{L}_i(QM^T e_i) = Q^T(-\mathsf{y}_i + \mathsf{p}_i),$$

where $\mathsf{p}_i \in \mathbb{R}^{K+1}$ and its $k$-th entry satisfying

$$\mathsf{p}_{ik} = \frac{\exp(e_k^T QM^T e_i)}{\sum_{k'=1}^{K+1} \exp(e_{k'}^T QM^T e_i)} \in [c, 1-c], \tag{S6.2}$$

for some constant $c \in (0, 1)$ depending on $(\tau, \alpha, K)$ only. Therefore,

$$\begin{aligned}
& n^{-1} \sum_{i=1}^{n} \nabla L_i(M^T e_i) \nabla L_i(M^T e_i)^T \\
&= n^{-1} Q^T \sum_{i=1}^{n} (\mathsf{y}_i - \mathsf{p}_i)(\mathsf{y}_i - \mathsf{p}_i)^T Q \\
&\succeq n^{-1} Q^T \sum_{l=1}^{\gamma n} \sum_{i \in S_l} (\mathsf{y}_i - \mathsf{p}_i)(\mathsf{y}_i - \mathsf{p}_i)^T Q \\
&\succeq n^{-1} Q^T \sum_{l \in L^*} \sum_{i \in S_l} (\mathsf{y}_i - \mathsf{p}_i)(\mathsf{y}_i - \mathsf{p}_i)^T Q \\
&:= n^{-1} Q^T \sum_{l \in L^*} A_l^T A_l Q,
\end{aligned}$$

where $A_l \in \mathbb{R}^{(K+1)\times(K+1)}$ has $K+1$ rows $\{\mathsf{y}_i - \mathsf{p}_i : i \in S_l\}$. We further note that $A_l$ is of the form $(I_{K+1} - \mathsf{P}_l)$ up to a rearrangement of the columns, where $\mathsf{P}_l \in \mathbb{R}^{(K+1)\times(K+1)}$ is a stochastic matrix with entries of the form

$$\frac{\exp(e_k^T QM^T e_i)}{\sum_{k'=1}^{K+1} \exp(e_{k'}^T QM^T e_i)}, \qquad i \in S_l, \quad k \in [K+1].$$

By (S6.2), for each $l \in L^*$, the stochastic matrix $\mathsf{P}_l$ is irreducible and aperiodic and $\ker(I_{K+1} - \mathsf{P}_l)$ is the span of the all-ones vector $\mathbf{1}$. Therefore,

$$(I_{K+1} - \mathsf{P}_l)QQ^T = (I_{K+1} - \mathsf{P}_l)(I_{K+1} - \tfrac{\mathbf{1}\mathbf{1}^T}{K+1}) = (I_{K+1} - \mathsf{P}_l).$$

It follows that

$$K = \operatorname{rank}(I_{K+1} - \mathsf{P}_l) = \operatorname{rank}((I_{K+1} - \mathsf{P}_l)QQ^T) \le \operatorname{rank}((I_{K+1} - \mathsf{P}_l)Q) \le K.$$

We conclude that the rank of $(I_{K+1} - \mathsf{P}_l)Q$ is $K$.

If $\mathcal{P}$ denotes the set of matrices $\{\mathsf{P} \in \mathbb{R}^{(K+1) \times (K+1)} : \text{stochastic with entries in } [c, 1-c]\}$, and $S^{K-1} = \{a \in \mathbb{R}^K : \|a\| = 1\}$. By compactness of $\mathcal{P}$ and $S^{K-1}$, we obtain

$$\frac{1}{n} \lambda_{\min}(\sum_{i=1}^n \nabla L_i(M^T e_i) \nabla L_i(M^T e_i)^T)$$

$$\geq \frac{1}{n} \sum_{l \in L^*} \lambda_{\min}(Q^T A_l^T A_l Q)$$

$$\geq \frac{1}{n} \sum_{l \in L^*} \min_{a \in S^{K-1}} a^T Q^T A_l^T A_l Q a$$

$$\geq \frac{1}{n} |L^*| \min_{a \in S^{K-1}, \mathsf{P} \in \mathcal{P}} a^T Q^T (I_{K+1} - \mathsf{P})^T (I_{K+1} - \mathsf{P}) Q a$$

$$\geq \frac{\gamma}{2} a_*^T Q^T (I_{K+1} - \mathsf{P}_*)^T (I_{K+1} - \mathsf{P}_*) Q a_*$$

$$\geq \frac{\gamma}{2} a_*^T Q^T (I_{K+1} - \mathsf{P}_*)^T (I_{K+1} - \mathsf{P}_*) Q a_*$$

$$:= m_*,$$

where $a_* \in S^{K-1}$, $\mathsf{P}_* \in \mathcal{P}$, and $m_*$ is a positive constant depending on $(K, \gamma, \tau)$ only. The first inequality above uses the property $\lambda_{\min}(A + B) \geq \lambda_{\min}(A) + \lambda_{\min}(B)$, where $A$ and $B$ are two positive semi-definite matrices.

In other words, $\frac{1}{n} \sum_{i=1}^n \nabla L_i(M^T e_i) \nabla L_i(M^T e_i)^T \succeq m_* I_K$. For the upper bound, since $\|Q\|_{op} \leq 1$ by definition of $Q$ and all the entries of $(\mathsf{y}_i - \mathsf{p}_i)(\mathsf{y}_i - \mathsf{p}_i)^T$ are between $-1$ and $1$ if Assumption 2.2 or Assumption S1.1 holds, we have

$$\|\sum_{i=1}^n \nabla L_i(M^T e_i) \nabla L_i(M^T e_i)^T\|_{op} = \|Q^T \sum_{i=1}^n (\mathsf{y}_i - \mathsf{p}_i)(\mathsf{y}_i - \mathsf{p}_i)^T Q\|_{op} \leq nK.$$

$\square$

**Lemma S6.2** (deterministic result on Hessian). *Let $L_i$ be defined as in* (S6.1). *For all $i \in [n]$, we have $\nabla^2 L_i(u) \preceq I_K$ for any $u \in \mathbb{R}^K$ and*

$$\min_{u \in \mathbb{R}^K, \|Qu\|_\infty \leq r} \nabla^2 L_i(u) \succeq c_* I_K,$$

*where $c_*$ is a positive constant depending on $(K, r)$ only.*

*Proof of Lemma S6.2.* Recall that $L_i(u) = \mathsf{L}_i(Qu)$, we have

$$\nabla^2 L_i(u) = Q^T \nabla^2 \mathsf{L}_i(Qu) Q,$$

where $\nabla^2 \mathsf{L}_i(Qu) = \text{diag}(\mathsf{p}_i) - \mathsf{p}_i \mathsf{p}_i^T$ and the $k$-th entry of $\mathsf{p}_i \in \mathbb{R}^{K+1}$ is defined as $\mathsf{p}_{ik} = \frac{\exp(e_k^T Qu)}{\sum_{k'=1}^{K+1} \exp(e_{k'}^T Qu)}$ for all $i \in [n], k \in [K+1]$. Thus, $\mathsf{p}_{ik} \leq 1$ for all $i \in [n], k \in [K+1]$. For any vector $a \in S^{K-1}$, we have

$$a^T \nabla^2 L_i(u) a = a^T Q^T [\text{diag}(\mathsf{p}_i) - \mathsf{p}_i \mathsf{p}_i^T] Q a$$

$$\leq a^T Q^T \text{diag}(\mathsf{p}_i) Q a$$

$$\leq \|Q^T \text{diag}(\mathsf{p}_i) Q\|_{op}$$

$$\leq 1,$$

where the last inequality uses $\|Q\|_{op} \leq 1$ and $\mathsf{p}_{ik} \leq 1$ for any $k$. Hence, $\nabla^2 L_i(u) \preceq I_K$ for any $i \in [n]$.

Now we prove the lower bound. For any $u \in \mathbb{R}^K$ such that $\|Qu\|_\infty \leq r$, we have

$$\mathsf{p}_{ik} = \frac{\exp(e_k^T Qu)}{\sum_{k'=1}^{K+1} \exp(e_{k'}^T Qu)} \in [c, 1-c]$$

for some constant $c$ depending on $(K, r)$ only. For any vector $a \in S^{K-1}$, let $\eta = Qa \in \mathbb{R}^{K+1}$, then $\mathbf{1}^T \eta = 0$ and

$$
\begin{aligned}
a^T \left[ \nabla^2 L_i(u) \right] a &= a^T Q^T [\mathrm{diag}(\mathsf{p}_i) - \mathsf{p}_i \mathsf{p}_i^T] Q a \\
&= \eta^T [\mathrm{diag}(\mathsf{p}_i) - \mathsf{p}_i \mathsf{p}_i^T] \eta \\
&= \sum_{k=1}^{K+1} \mathsf{p}_{ik} \eta_k^2 - \Big( \sum_{k=1}^{K+1} \mathsf{p}_{ik} \eta_k \Big)^2 \\
&> \sum_k \mathsf{p}_{ik} \eta_k^2 - \sum_k \mathsf{p}_{ik} \eta_k^2 \sum_k \mathsf{p}_{ik} \\
&= \sum_k \mathsf{p}_{ik} \eta_k^2 (1 - \sum_k \mathsf{p}_{ik}) \\
&= 0,
\end{aligned}
$$

where the last equality uses $\sum_{k=1}^{K+1} \mathsf{p}_{ik} = 1$, and the inequality follows by $(\sum_k \mathsf{p}_{ik} \eta_k)^2 = (\sum_k \sqrt{\mathsf{p}_{ik}} \sqrt{\mathsf{p}_{ik}} \eta_k)^2 \leq \sum_k \mathsf{p}_{ik} \sum_k \mathsf{p}_{ik} \eta_k^2$ using the Cauchy-Schwarz inequality, and here " $=$ " holds if and only if $\sqrt{\mathsf{p}_{ik}} \propto \sqrt{\mathsf{p}_{ik}} \eta_k$ for each $k$, which is not true since $\mathsf{p}_{ik} \in [c, 1-c]$ and $\mathbf{1}^T \eta = 0$.

Let $\mathcal{H} = \{ Q^T (\mathrm{diag}(\mathsf{p}) - \mathsf{p}\mathsf{p}^T) Q : \mathsf{p} \in [c, 1-c]^{K+1} \}$, then $\mathcal{H}$ is compact and

$$
\min_{u \in \mathbb{R}^K, \|Qu\|_\infty \leq r} \lambda_{\min}(\nabla^2 L_i(u)) \geq \min_{a \in S^{K-1}, H \in \mathcal{H}} a^T H a = a_*^T H_* a_* > 0
$$

for some $a_* \in S^{K-1}$ and $H_* \in \mathcal{H}$. Therefore,

$$
\min_{u \in \mathbb{R}^K, \|Qu\|_\infty \leq r} \nabla^2 L_i(u)) \succeq c_* I_K
$$

where $c_*$ is a positive constant depending on $(K, r)$ only. $\qquad \square$

## S6.2 Lipschitz conditions

We first restate the definitions of following sets,

$$
U = \Big\{ (Y, X) \in \mathbb{R}^{n \times (K+1)} \times \mathbb{R}^{n \times p} : \hat{\mathsf{B}} \text{ exists}, \|X\hat{\mathsf{B}}(I_{K+1} - \tfrac{\mathbf{1}\mathbf{1}^T}{K+1})\|_F^2 < n\tau \Big\},
$$

$$
U_y = \Big\{ Y \in \mathbb{R}^{n \times (K+1)} : \sum_{i=1}^n I(\mathsf{y}_{ik} = 1) \geq \gamma n \text{ for all } k \in [K+1] \Big\}
$$

**Lemma S6.3.** *Assume $p/n \leq \delta^{-1} < 1 - \alpha$ for some $\alpha \in (0, 1)$ and $\delta > 1$. Let $\mathcal{I} = \{ I \subset [n] : |I| = \lceil n(1-\alpha) \rceil \}$ and $P_I = \sum_{i \in I} e_i e_i^T$. Define*

$$
U_x = \Big\{ X \in \mathbb{R}^{n \times p} : \min_{I \in \mathcal{I}} \lambda_{\min} \big( \tfrac{\Sigma^{-1/2} X^T P_I X \Sigma^{-1/2}}{n} \big) \geq \phi_*^2, \tfrac{\|X\Sigma^{-1/2}\|_{op}}{\sqrt{n}} \leq \phi^* \Big\} \qquad \text{(S6.3)}
$$

*for some positive constants $\phi_*, \phi^*$, which depend on $(\delta, \alpha)$ only. Let $U^* = \{ (Y, X) \in U : Y \in U_y, X \in U_x \}$. Then under Assumptions 2.1, 2.3 and 2.4, and if either Assumption 2.2 or Assumption S1.1 holds, we have*

(i) $\mathbb{P}((Y, X) \in U^*) \to 1$ *as $n, p \to \infty$.*

(ii) *Let $G$ be defined in Assumption S5.2. If $\{(Y, X), (Y, \tilde{X})\} \subset U^*$, we have*

$$
\|G(Y, X) - G(Y, \tilde{X})\|_F \leq L\|(X - \tilde{X})\Sigma^{-1/2}\|_F,
$$

*where $L$ is a positive constant depending on $(K, \gamma, \tau, \alpha)$ only.*

*Proof of Lemma S6.3.* We first prove statement (i). Under Assumption 2.1, [Bellec, 2022, Lemma 7.7] implies

$$
\mathbb{P}\big( \min_{I \in \mathcal{I}} \lambda_{\min} \big( \tfrac{\Sigma^{-1/2} X^T P_I X \Sigma^{-1/2}}{n} \big) \geq \phi_*^2 \big) \to 1
$$

for some positive constant $\phi_*$ depending on $(\delta, \alpha)$ only. Furthermore, [Davidson and Szarek, 2001, Theorem II.13] implies

$$\mathbb{P}\left(\frac{\|X\Sigma^{-1/2}\|_{op}}{\sqrt{n}} \leq \phi^*\right) \to 1$$

for some positive constant $\phi^*$ depending on $\delta$ only. Therefore, $\mathbb{P}(X \in U_x) \to 1$. Under Assumption 2.3, we have $\mathbb{P}(Y \in U_y) \to 1$. Under Assumption 2.4, we have $\mathbb{P}((Y, X) \in U) \to 1$. In conclusion, under Assumptions 2.1, 2.3 and 2.4, we have $\mathbb{P}((Y, X) \in U^*) \to 1$ as $n, p \to \infty$.

Now we prove the statement (ii). For a fixed $Y$, let $(Y, X), (Y, \tilde{X}) \in U^*$, $\hat{B}, \tilde{B}$ be their corresponding minimizers of (S3.3), and $G, \tilde{G}$ be their corresponding gradient matrices. We first provide some useful results derived from the KKT conditions. From the KKT conditions $X^T G = \tilde{X}^T \tilde{G} = 0$, we have

$$
\begin{aligned}
\langle X\hat{B} - \tilde{X}\tilde{B}, G - \tilde{G} \rangle =& \langle \hat{B} - \tilde{B}, \tilde{X}^T \tilde{G} - X^T G \rangle + \langle X\hat{B} - \tilde{X}\tilde{B}, G - \tilde{G} \rangle \\
=& -\langle (X - \tilde{X})(\hat{B} - \tilde{B}), G \rangle + \langle (X - \tilde{X})\hat{B}, G - \tilde{G} \rangle.
\end{aligned}
$$

Since $\|\nabla^2 L_i(u)\|_{op} \leq 1$ for any $u \in \mathbb{R}^K$ from Lemma S6.2, $\nabla L_i(\cdot)$ is 1-Lipschitz. Thus,

$$
\begin{aligned}
\langle X\hat{B} - \tilde{X}\tilde{B}, G - \tilde{G} \rangle =& \sum_{i=1}^n \langle \hat{B}^T x_i - \tilde{B}^T \tilde{x}_i, \nabla L_i(\hat{B}^T x_i) - \nabla L_i(\tilde{B}^T \tilde{x}_i) \rangle \\
\geq& \sum_{i=1}^n \langle \nabla L_i(\hat{B}^T x_i) - \nabla L_i(\tilde{B}^T \tilde{x}_i), \nabla L_i(\hat{B}^T x_i) - \nabla L_i(\tilde{B}^T \tilde{x}_i) \rangle \\
=& \|G - \tilde{G}\|_F^2.
\end{aligned}
$$

If $(Y, X), (Y, \tilde{X}) \in U$, we have $\|X\hat{B}Q^T\|_F^2 + \|\tilde{X}\tilde{B}Q^T\|_F^2 \leq 2n\tau$. That is,

$$\sum_{i=1}^n \left(\|Q\hat{B}^T x_i\|^2 + \|Q\tilde{B}^T \tilde{x}_i\|^2\right) \leq 2n\tau.$$

Define the index set

$$I = \{i \in [n] : \|Q\hat{B}^T x_i\|^2 + \|Q\tilde{B}^T \tilde{x}_i\|^2 \leq \tfrac{2\tau}{\alpha}\},$$

then we have $|I| \geq (1 - \alpha)n$ by Markov's inequality. Thus, for all $i \in I$, we have $\|Q\hat{B}^T x_i\|_\infty \vee \|Q\tilde{B}^T \tilde{x}_i\|_\infty \leq \sqrt{\tfrac{2\tau}{\alpha}}$.

Applying Lemma S6.2 with $r = \sqrt{\tfrac{2\tau}{\alpha}}$ gives

$$\min_{\|Qu\|_\infty \leq \sqrt{\frac{2\tau}{\alpha}}} \nabla^2 L_i(u) \succeq c_* I_K,$$

where $c_*$ is a constant depending on $(K, \tau, \alpha)$. Therefore,

$$
\begin{aligned}
c_* \|P_I(X\hat{B} - \tilde{X}\tilde{B})\|_F^2 =& c_* \sum_{i \in I} \|\hat{B}^T x_i - \tilde{B}^T \tilde{x}_i\|^2 \\
\leq& \sum_{i \in I} \langle \hat{B}^T x_i - \tilde{B}^T \tilde{x}_i, \nabla L_i(\hat{B}^T x_i) - \nabla L_i(\tilde{B}^T \tilde{x}_i) \rangle \\
\leq& \sum_{i=1}^n \langle \hat{B}^T x_i - \tilde{B}^T \tilde{x}_i, \nabla L_i(\hat{B}^T x_i) - \nabla L_i(\tilde{B}^T \tilde{x}_i) \rangle \\
=& \langle X\hat{B} - \tilde{X}\tilde{B}, G - \tilde{G} \rangle \\
=& -\langle (X - \tilde{X})(\hat{B} - \tilde{B}), G \rangle + \langle (X - \tilde{X})\hat{B}, G - \tilde{G} \rangle.
\end{aligned}
$$

We next bound the first line from below by expanding the squares,

$$
\begin{aligned}
\|P_I(X\hat{B} - \tilde{X}\tilde{B})\|_F^2 =& \|P_I\tilde{X}(\hat{B} - \tilde{B}) + P_I(X - \tilde{X})\hat{B}\|_F^2 \\
\geq& \|P_I\tilde{X}(\hat{B} - \tilde{B})\|_F^2 + 2\langle P_I\tilde{X}(\hat{B} - \tilde{B}), P_I(X - \tilde{X})\hat{B} \rangle \\
\geq& n\phi_*^2 \|\Sigma^{1/2}(\hat{B} - \tilde{B})\|_F^2 + 2\langle \tilde{X}(\hat{B} - \tilde{B}), P_I(X - \tilde{X})\hat{B} \rangle,
\end{aligned}
$$

where in the last inequality we use the constant $\phi^*$ in (S6.3). Therefore, we obtain

$$c_*\phi_*^2 n\|\Sigma^{1/2}(\hat{B}-\tilde{B})\|_F^2$$

$$\leq -\langle(X-\tilde{X})(\hat{B}-\tilde{B}),G\rangle + \langle(X-\tilde{X})\hat{B},G-\tilde{G}\rangle - 2c_*\langle \tilde{X}(\hat{B}-\tilde{B}),P_I(X-\tilde{X})\hat{B}\rangle.$$

Together with the inequality that $\|G-\tilde{G}\|_F^2 \leq \langle X\hat{B}-\tilde{X}\tilde{B},G-\tilde{G}\rangle$, we obtain

$$c_*\phi_*^2 n\|\Sigma^{1/2}(\hat{B}-\tilde{B})\|_F^2 + \|G-\tilde{G}\|_F^2$$

$$\leq -2\langle(X-\tilde{X})(\hat{B}-\tilde{B}),G\rangle + 2\langle(X-\tilde{X})\hat{B},G-\tilde{G}\rangle - 2c_*\langle \tilde{X}(\hat{B}-\tilde{B}),P_I(X-\tilde{X})\hat{B}\rangle$$

$$\leq (4+2c_*\phi^*)\|(X-\tilde{X})\Sigma^{-1/2}\|_{op}\big(\|\Sigma^{1/2}(\hat{B}-\tilde{B})\|_F \vee \frac{\|G-\tilde{G}\|_F}{\sqrt{n}}\big)\big(\|\Sigma^{1/2}\hat{B}\|_F \vee \frac{\|G\|_{op}}{\sqrt{n}}\big)\sqrt{n},$$

where we bound $\langle \tilde{X}(\hat{B}-\tilde{B}),P_I(X-\tilde{X})\hat{B}\rangle$ by definition of $\phi^*$,

$$\langle \tilde{X}(\hat{B}-\tilde{B}),P_I(X-\tilde{X})\hat{B}\rangle$$

$$=\langle \Sigma^{1/2}(\hat{B}-\tilde{B}),\Sigma^{-1/2}\tilde{X}^T P_I(X-\tilde{X})\hat{B}\rangle$$

$$\leq \|\Sigma^{1/2}(\hat{B}-\tilde{B})\|_F\|P_I\tilde{X}\Sigma^{-1/2}\|_{op}\|(X-\tilde{X})\Sigma^{-1/2}\|_{op}\|\Sigma^{1/2}\hat{B}\|_F$$

$$\leq \sqrt{n}\phi^*\|\Sigma^{1/2}(\hat{B}-\tilde{B})\|_F\|(X-\tilde{X})\Sigma^{-1/2}\|_{op}\|\Sigma^{1/2}\hat{B}\|_F.$$

Now we derive a bound of the form $\|\Sigma^{1/2}\hat{B}\|_F \lesssim \|G\|_F/\sqrt{n}$. To this end, since $\phi_*\|\Sigma^{1/2}\hat{B}\|_F \leq \|P_I X\hat{B}\|_F/\sqrt{n} \leq \|X\hat{B}\|_F/\sqrt{n} = \|X\hat{B}Q^T\|_F/\sqrt{n} \leq \sqrt{\tau}$.

Applying Lemma S6.1 to $M = X\hat{B}$, we have $\frac{1}{n}\sum_{i=1}^n g_i g_i^T \succeq m_* I_K$. Therefore,

$$\frac{1}{n}\|G\|_F^2 = \frac{1}{n}\sum_{i=1}^n \|g_i\|^2 = \frac{1}{n}\sum_{i=1}^n \mathrm{Tr}(g_i g_i^T) \geq K m_*.$$

This implies that

$$\phi_*^2\|\Sigma^{1/2}\hat{B}\|_F^2 \leq \tau \leq \frac{\tau}{K m_*(I)}\|G\|_F^2/n.$$

In conclusion, if $\{(Y,X),(Y,\tilde{X})\} \subset U^*$ then

$$\sqrt{n}\|\Sigma^{1/2}(\hat{B}-\tilde{B})\|_F + \|G-\tilde{G}\|_F \leq C n^{-1/2}\|(X-\tilde{X})\Sigma^{-1/2}\|_{op}\|G\|_F \leq CK\|(X-\tilde{X})\Sigma^{-1/2}\|_{op},$$

where $C$ is a constant depending on $(K,\gamma,\tau,\alpha)$ only. Note that $\|G\|_F \leq \sqrt{nK}$ since all entries of $G$ are in $[-1,1]$. $\qquad\square$

**Lemma S6.4.** *If $p/n \leq \delta^{-1} < (1-\alpha)$ for some $\alpha \in (0,1)$ and $\delta > 1$. If $(Y,X) \in U$ and $X \in U_x$, where $U_x$ is defined in Lemma S6.3, we have*

$$\frac{1}{n}\sum_{i=1}^n H_i \otimes (x_i x_i^T) \succeq c_1(I_K \otimes \Sigma),$$

*where $c_1$ is a positive constant depending on $(K,\tau,\alpha,\phi_*)$ only.*

*Proof of Lemma S6.4.* If $(Y,X) \in U$, we have $\|X\hat{B}Q^T\|_F^2 \leq n\tau$. Define the index set

$$I = \{i \in [n] : \|Q\hat{B}^T x_i\| \leq \frac{\tau}{\alpha}\},$$

then we have $|I| \geq (1-\alpha)n$ by Markov's inequality. Therefore, for any $i \in I$, $\|Q\hat{B}^T x_i\|_\infty \leq \frac{\tau}{\alpha}$. Applying Lemma S6.2 with $u = \hat{B}^T x_i$ and $r = \frac{\tau}{\alpha}$, we have for any $i \in I$, $H_i = \nabla^2 L_i(\hat{B}^T x_i) \succeq c_* I_K$ for some positive constant $c_*$ depending on $(K,\tau,\alpha)$ only. Therefore, if $(Y,X) \in U$ and $X \in U_x$, we have

$$\frac{1}{n}\sum_{i=1}^n H_i \otimes (x_i x_i^T) \succeq \frac{1}{n}\sum_{i\in I} H_i \otimes (x_i x_i^T)$$

$$\succeq c_*\frac{1}{n}\sum_{i\in I} I_K \otimes (x_i x_i^T)$$

$$= c_*(I_K \otimes \frac{X^T P_I X}{n})$$

$$\succeq c_*\phi_*(I_K \otimes \Sigma)$$

$$= c_1(I_K \otimes \Sigma),$$

where $P_I = \sum_{i \in I} e_i e_i^T$ and $c_1$ is a positive constant depending on $(K, \tau, \alpha, \phi_*)$ only. $\qquad\square$

### S6.3 Proof of Theorem S3.1

The proof of Theorem S3.1 is a direct consequence of Theorem S5.1 by noting

$$\sqrt{n}\Omega_{jj}^{-1/2}\Big(\frac{1}{n}\sum_{i=1}^n g_i g_i^T\Big)^{-1/2}\Big(\frac{1}{n}\sum_{i=1}^n V_i\Big)\hat{B}^T e_j = \Omega_{jj}^{-1/2}(G^T G)^{-1/2} V \hat{B}^T e_j,$$

which is a consequence of the identities $G = \sum_{i=1}^n e_i g_i^T$ and $V = \sum_{i=1}^n V_i$.

It thus remains to verify the conditions (1) and (2) in Theorem S5.1 from the assumptions in Theorem S3.1.

Applying Lemma S6.3 with $\alpha$ chosen as $1 - \delta^{-1/2}$, we have for $\{(Y, X), (Y, \tilde{X})\} \subset U^*$,

$$\|G(Y, X) - G(Y, \tilde{X})\|_F \leq L\|(X - \tilde{X})\Sigma^{-1/2}\|_F,$$

where $L$ is a positive constant depending on $(K, \gamma, \tau, \delta)$ only.

Apply Lemma S6.4 with the same $\alpha = 1 - \delta^{-1/2}$, we have for $(Y, X) \in U^*$,

$$\frac{1}{n}\sum_{i=1}^n H_i \otimes (x_i x_i^T) \succeq c_*(I_K \otimes \Sigma),$$

where $c_*$ is a positive constant depending on $(K, \tau, \delta)$ only.

Applying Lemma S6.1 with $M = X\hat{B}$, we have for $(Y, X) \in U^*$,

$$m_* I_K \preceq n^{-1}\sum_{i=1}^n \nabla L_i(M^T e_i)\nabla L_i(M^T e_i)^T \preceq K I_K,$$

where $m_*$ is a positive constant depending on $(K, \gamma, \tau)$ only. Therefore, the conditions (1) and (2) in Theorem S5.1 hold when the multinomial logistic loss is used. This completes the proof of Theorem S3.1.

## S7 Other proof

### S7.1 Proof of Equation (1.12) (Classical asymptotic theory with fixed $p$)

Here we provide a derivation of the asymptotic distribution of MLE under classical setting, where $p$ is fixed and $n$ tends to infinity.

We first calculate the Fisher information matrix of the multinomial logistic log-odds model (1.8) with covariate $x \sim N(\mathbf{0}, \Sigma)$ and response $y \in \{0, 1\}^{K+1}$ one-hot encoded satisfying $\sum_{k=1}^{K+1} y_k = 1$. Note that the model (1.8) can be rewritten as

$$\mathbb{P}(y_k = 1|x) = \frac{\exp(x^T A^* e_k)}{1 + \sum_{k'=1}^K \exp(x^T A^* e_{k'})}, \quad \forall k \in \{1, ..., K\}$$

$$\mathbb{P}(y_{K+1} = 1|x) = \frac{1}{1 + \sum_{k'=1}^K \exp(x^T A^* e_{k'})}.$$

The likelihood function of a parameter $A \in \mathbb{R}^{p \times K}$ is

$$L(A) = \prod_{k=1}^K \Big[\frac{\exp(x^T A e_k)}{1 + \sum_{k'=1}^K \exp(x^T A e_{k'})}\Big]^{y_k} \Big[\frac{1}{1 + \sum_{k'=1}^K \exp(x^T A e_{k'})}\Big]^{y_{K+1}}$$

$$= \prod_{k=1}^K [\exp(x^T A e_k)]^{y_k} \frac{1}{1 + \sum_{k'=1}^K \exp(x^T A e_{k'})},$$

where we used $\sum_{k=1}^{K+1} \mathsf{y}_k = 1$. Thus, the log-likelihood function is

$$\ell(A) = \sum_{k=1}^{K} \mathsf{y}_k x^T A e_k - \log\Big[1 + \sum_{k'=1}^{K} \exp(x^T A e_{k'})\Big].$$

It is more convenient to calculate the Fisher information matrix on the vector space $\mathbb{R}^{pK}$ instead of the matrix space $\mathbb{R}^{p \times K}$. To this end, let $\theta = \mathrm{vec}(A^T)$, then $x^T A e_k = e_k^T A^T x = (x^T \otimes e_k^T)\,\mathrm{vec}(A^T) = (x^T \otimes e_k^T)\theta$, and the log-likelihood function parameterized by $\theta$ is

$$\ell(\theta) = \sum_{k=1}^{K} y_k (x^T \otimes e_k^T)\theta - \log[1 + \sum_{k'=1}^{K} \exp((x^T \otimes e_{k'}^T)\theta)].$$

By multivariate calculus, we obtain the Fisher information matrix evaluated at $\theta^* = \mathrm{vec}(A^{*T})$,

$$\begin{aligned}
\mathcal{I}(\theta^*) &= -\mathbb{E}\Big[\frac{\partial}{\partial\theta}\frac{\partial\ell(\theta)}{\partial\theta^T}\Big]\Big|_{\theta=\theta^*} \\
&= \mathbb{E}[(xx^T) \otimes (\mathrm{diag}(\pi^*) - \pi^*\pi^{*T})],
\end{aligned}$$

where $\pi^* \in \mathbb{R}^K$ with $k$-th entry $\pi_k^* = \frac{\exp(x^T A^* e_k)}{1+\sum_{k'=1}^{K}\exp(x^T A^* e_{k'})}$ for each $k \in [K]$.

From classical maximum likelihood theory, for instance [Van der Vaart, 1998, Chapter 5], we have

$$\sqrt{n}(\hat{\theta} - \theta^*) \overset{d}{\longrightarrow} N(\mathbf{0}, \mathcal{I}_{\theta^*}^{-1}),$$

where $\hat{\theta} = \mathrm{vec}(\hat{A}^T)$ and $\hat{A}$ is the MLE of $A^*$. Furthermore, if the $j$-th covariate is independent of the response, we know $e_j^T A^* = \mathbf{0}^T$, then

$$\sqrt{n}\hat{A}^T e_j = \sqrt{n}(\hat{A}^T e_j - A^{*T} e_j) = \sqrt{n}(e_j^T \otimes I_K)(\hat{\theta} - \theta^*) \overset{d}{\longrightarrow} N(\mathbf{0}, S_j),$$

where

$$\begin{aligned}
S_j &= (e_j^T \otimes I_K)\mathcal{I}_{\theta^*}^{-1}(e_j \otimes I_K) \\
&= e_j^T \mathrm{cov}(x)^{-1} e_j [\mathbb{E}(\mathrm{diag}(\pi^*) - \pi^*\pi^{*T})]^{-1}
\end{aligned}$$

holds by the independence between the $j$-th covariate and the response under $H_0$ in (1.10). This completes the proof of Equation (1.12).

