# OpenReview forum: "Multinomial Logistic Regression: Asymptotic Normality on Null Covariates in High-Dimensions"
_NeurIPS.cc/2023/Conference — NeurIPS 2023 poster_

### Official Review · Reviewer_U9Cr · 2023-07-04

**Soundness:** 3 good
**Presentation:** 4 excellent
**Contribution:** 3 good
**Rating:** 7
**Confidence:** 3

**Summary:**

The paper aims to study the statistics of the multinomial logistic MLE on null covariates in a multiclass classification setting in the asymptotic proportional regime, i.e., in the limit in which the size of the dataset $n$ and of the dimension of the covariates $p$ are both sent to infinity but with finite ratio. The work extends previous results by Sur and Candès in which the normality of the MLE on null covariates was proven in the case of binary classification and introduces a proper statistic for a $\chi^2$ test on the relevance of a feature.

**Strengths:**

The paper is clearly and carefully written, the validity conditions of the results are clearly stated and the outcome of the theoretical analysis is supported by robust numerical evidence. The characterization of the statistics of the MLE on null covariates in the case of multiclass classification in the high dimensional limit is timely. The authors also compare the high-d and the classical theory, showing the remarkable difference in the provided predictions. Finally, they also test the validity of their results beyond the Gaussian hypothesis adopted for the derivation of their theorem.

**Weaknesses:**

A possible theoretical weakness of the paper is the fact that the existence of the MLE is assumed (Assumption 2.4) and not fully characterized. This "weakness" is acknowledged by the authors themselves in the final section of the main text.

**Questions:**

Out of curiosity, the authors state that they expect their results to hold for covariates with distribution having "sufficiently light" tails. Could they comment on this? Do they expect, for example, sub-gaussianity to be sufficient for their results to hold?

As a (very) minor remark, Theorem 2.1 and Theorem 2.2 exhibit the quantities $\mathsf y_i-\hat{\mathsf p}_i\equiv -\mathsf g_i$, but a different notation is used. The analogy of the two results might appear more evident at first sight using the same notation in both theorems (eg replacing $-\mathsf y_i+\hat{\mathsf p}_i$ with $\mathsf g_i$ in the first theorem).

**Limitations:**

The work is of theoretical nature and all limitations are within the well specified hypotheses of the reported theorems.

---

> ### Author Rebuttal · Authors · 2023-08-06
>
> Thank you for the insightful and encouraging feedback. Here, we respond to your comments point by point.
>
> > **Q1:** Out of curiosity, the authors state that they expect their results to hold for covariates with distribution having "sufficiently light" tails. Could they comment on this? Do they expect, for example, sub-gaussianity to be sufficient for their results to hold?
>
> **A1:**
> This is a great question. Regarding this question of universality with respect to the distribution of $x_i$, the associated challenges and some recent references, we refer to our answer to question **Q2** from Reviewer uvEK.
>
> > **Q2:** As a (very) minor remark, Theorem 2.1 and Theorem 2.2 exhibit the quantities $\mathsf{y}_i - \mathsf{\hat p}_i = -\mathsf{g}_i$
> , but a different notation is used. The analogy of the two results might appear more evident at first sight using the same notation in both theorems (eg replacing $-\mathsf{y}_i + \mathsf{\hat p}_i$ with $\mathsf{g}_i$ in the first theorem).
>
> **A2:**
> Thank you very much for your attention to detail.
> We adopt your suggestion and revised the manuscript by replacing
> $-\mathsf{y}_i + \mathsf{\hat p}_i$ with $\mathsf{g}_i$ in Theorem 2.1 to make the analogy between the two results more evident.

---

### Official Review · Reviewer_uvEK · 2023-07-06

**Soundness:** 3 good
**Presentation:** 3 good
**Contribution:** 3 good
**Rating:** 7
**Confidence:** 2

**Summary:**

This paper studies the asymptotic distribution of multinomial logistic regression in the high dimensions. Under the assumption of a Gaussian design (and a few other assumptions), it establishes and characterizes the asymptotic normality of the coefficient of a null feature. This extends previous results for high-dimensional binary logistic regression to $K \geq 3$ number of classes. This result enables proper significant testing in high-dimensions, for which the classical fixed-$p$ asymptotic fails to control the type-I error.

**Strengths:**

1. The paper extends previous results for high-dimensional logistic regression to the multi-class setting.
2. The paper is well-written and the technical content is presented rigorously.
3. The simulations results seem to match the asymptotic theory very well.

**Weaknesses:**

1. The hypothesis $H_0$ seems stronger than the hypothesis $H_0'$: the population coefficient (in terms of KL projection) for feature $j$ is zero. Supposedly $H_0'$ reduces to $H_0$ when the model is well-specified. In terms of the data generating mechanism given by Assumption 2.1, it seems possible that when $y_i = f(U_i, x_i^{T} B^{\ast})$ does not hold, a feature that does not satisfy $H_0$ may still have zero as its population "true" coefficient. I am wondering whether $H_0$ can be suitably weakened.
2. The result relies on a random Gaussian design, although it is suggested that the result probably extends to other random designs with light tails.
3. The paper does not characterize the distribution for non-null features, so the study of parameter inference for high-dimensional multinomial logistic regression is still incomplete.

**Questions:**

1. Per my first comment on the "Weaknesses", can authors comment on the extent to which $H_0$ can be weakened?
2. To what extent is the "universality" expected to hold for this problem? Does the asymptotic normality fail under a random, heavy-tailed design? What about fixed designs?

**Limitations:**

I do not foresee any potential negative societal impact of this work. In terms of limitations, I think obtaining results for the non-null features would make this work a much stronger paper.

---

> ### Author Rebuttal · Authors · 2023-08-06
>
> Thank you for your insightful feedback to help us improve our paper. Here, we respond to your comments point by point.
>
> > **Q1**: Per my first comment on the "Weaknesses", can authors comment on the extent to which can be weakened?
>
> **A1**: The question regarding the possibility to weaken $H_0$ to $H_0'$ is interesting. We do not have an answer right away. The proof is written with a generative model of the form $y_i = f(U_i, x_i^T B^*)$ in mind, and it is not clear at this point that all steps in the proof where this model is used could be generalized to a model-free setting. (Note that many choices for $f$ are allowed which provides significant model flexibility; the stronger assumption here is indeed that the response depends on $x_i$ only through $x_i^TB^*$).
>
> > **Q2**: To what extent is the "universality" expected to hold for this problem? Does the asymptotic normality fail under a random, heavy-tailed design? What about fixed designs?
>
> **A2**: Thanks for the great question. We expect our asymptotic normality results (Theorems 2.1 and 2.2) to hold when the covariate $x_i$ is iid sub-Gaussian or has bounded moments of sufficiently large order. This is expected because
>
> 1. This is empirically confirmed by the simulation results using the Radermacher distribution and SNP example in Section 3. However, the asymptotical normality fails to hold under random, heavy-tailed design; we conducted additional simulation results using heavy-tailed covariates (e.g., Pareto and Weibull); the corresponding quantile-quantile lines deviate significantly from the 45-degree line.
> 2. In the same regime (proportional asymptotics for p/n), several recent works show that universality holds under a set of assumptions that typically includes iid entries with bounded moments of sufficiently large order:
>     - "Universality of empirical risk minimization" by Montanari and Saheed (arxiv:2202.08832), cf. section 3.3 there for iid sub-gaussian entries,
>     - "Universality laws for Gaussian mixtures in generalized linear models" by Dandi, Stephan, Krzakala, Loureiro, Zdeborová (arxiv:2302.08933)
>     - "Universality of regularized regression estimators in high dimensions" by Han and Shen (arxiv:2206.07936) which handles iid entries with $6+\epsilon$ bounded moments.
>
>
> These three works provide hope for extending our results to iid entries (or other design distributions treated in these works). However, all these works assume some form of "delocalization" of $\hat B$ in the form of an $L_\infty$-norm constraint in the minimization problem (see, e.g., equation (29) in Montanari and Saheed). This is fine if $\hat B$ is already known to satisfy such $L_\infty$-norm bound, however in logistic regression (even in two classes) such $L_\infty$-norm bound is not known, and these universality results cannot yet be applied. Given the intricacies involved in proving the delocalization and universality results, we prefer to leave this problem open for future research as these challenging universality questions warrant their own papers.
>
> > I think obtaining results for the non-null features would make this work a much stronger paper.
>
> We agree that an analysis for non-null features would be very interesting. However,
> a major obstacle to obtaining such results is that we operate under the model
> $y_i=f(U_i,x_i^TB^*)$, allowing for a general function $f$ ($f$ must output a
> one-hot encoded vector, but is otherwise unrestricted).  In our setting, $B^*$
> is thus not identifiable because $y_i=\tilde f(U_i, x_i^T\tilde{B}^*)$, where
> $\tilde{B}^*=B^* M$ and $\tilde{f}(u,v^T)=f(u,v^TM^{-1})$ for any invertible matrix
> $M$.  Consequently, turning to confidence intervals for entries of $B^*$ or
> confidence sets for rows of $B^*$ would require either a model more specific
> than the current model of the paper (in order to address the identifiability
> issue), or considering that the parameter of interest is the column space of
> $B^*$. Both of these avenues appear interesting for future work. The rationale for
> leaving such extensions for future work is that they would require mathematics
> significantly different from the present work (mathematics relying on
> additional assumptions compared to our model $y_i=f(U_i,x_i^TB^*)$) and we are
> not aware of known techniques that could advance the analysis for non-null
> features.

---

> > ### Comment · Reviewer_uvEK · 2023-08-14
> >
> > I appreciate the reply from the authors. I feel that my concerns have been properly addressed. I am raising my score to "Accept".

---

### Official Review · Reviewer_7JUy · 2023-07-07

**Soundness:** 4 excellent
**Presentation:** 4 excellent
**Contribution:** 3 good
**Rating:** 7
**Confidence:** 4

**Summary:**

In this paper, the authors provide asymptotic characterization of the behavior of the maximum likelihood estimator (MLE) of multinomial logistic model (with more than two classes), in the high-dimensional regime where the dimension and the sample size of data go to infinity at the same rate.

Under some technical assumptions (that may need some further elaborations), this paper develops asymptotic normality and asymptotic chi-square results for the multinomial logistic MLE on null covariates (see Theorem 2.1 and 2.2). The proposed results can be used for statistical inference and significance test of some specific features. Numerical experiments on synthetic data are provided in Section 3 to validate the proposed theory.

**Strengths:**

This papers focuses on the fundamental and important problem of MLE of multinomial logistic model in the modern high-dimensional regime.
The proposed theory improves prior art in characterizing the asymptotic normality and asymptotic chi-square results, both of significance to statistics and ML.
The paper is in general well written and easy to follow.


**Weaknesses:**

I do not have strong concerns to raise for this paper.
See below for some detailed comments and/or questions.

**Questions:**

1. While I am almost fine with Assumption 2.1 and 2.2 (just being curious, is some upper bound on the spectral norm of the covariance $\Sigma$ needed? or it is just a matter of scaling with respect to $n,p$?), I am a bit confused by Assumption 2.3 and 2.4: are they something intrinsic or for the ease of technical analysis? What happens if, say Assumption 2.3 is violated? Can we have something similar but just more involved or the MLE is totally different? Also, Assumption 2.4 is a bit misleading, in the sense that the assumption is not instinct, and should perhaps be reduced into some assumption on the dimension ratio $p/n$ and/or statistics of the data? I believe it makes more sense to assume something like "the dimension ratio $p/n$, covariance $\Sigma$ and xxx satisfy that xxx". I am also confused by the paragraph after Assumption 2.4 and I am not sure the convergence of some multinomial regression solver can be used as a rigorous theoretical indicator. The algorithm may converge (or believed to converge) due to many reasons. Perhaps some better (numerical) criterion can be proposed by, e.g., checking the gradient and/or Hessian of the point of interest.
2. for the sake of presentation and use, it be helpful to present Theorem 2.2 and the estimation of $\Omega_{jj}$ in form of an algorithm.
3. Almost nothing is mentioned for the proof of the theoretical results (Theorem 2.1 and 2.2): is the proof technically challenging or contains some ingredients and/or intermediate results that may be of independent interest? Could the authors elaborate more on this?

**Limitations:**

This paper is primarily of a theoretical nature, and I do not see any potential negative societal impact of this work.

---

> ### Author Rebuttal · Authors · 2023-08-06
>
> Thank you for your valuable and encouraging feedback to help us improve our paper. Here we provide answers to your concerns point by point.
>
> > **Q1:** While I am almost fine with Assumption 2.1 and 2.2 (just being curious, is some upper bound on the spectral norm of the covariance $\Sigma$ needed? or it is just a matter of scaling with respect to $n,p$?), I am a bit confused by Assumption 2.3 and 2.4: are they something intrinsic or for the ease of technical analysis? What happens if, say Assumption 2.3 is violated? Can we have something similar but just more involved or the MLE is totally different? Also, Assumption 2.4 is a bit misleading, in the sense that the assumption is not instinct, and should perhaps be reduced into some assumption on the dimension ratio $p/n$ and/or statistics of the data? I believe it makes more sense to assume something like "the dimension ratio $p/n$, covariance $\Sigma$ and xxx satisfy that xxx".
>
> **A1:**
>
> The upper bound on the spectral norm of $\Sigma$ is not needed in our theory thanks to the rotational invariance: a key step in our proof involves rotating and scaling the covariate with covariance $\Sigma$ to the identity matrix $I_p$ (see page 20 in the supplement).
>
> Assumption 2.3 is indeed necessary for our technical analysis. One intuition that this is a fairly mild assumption is that if for some label $k_0$ we observe $o(n)$ observations, the dataset obtained by dropping all observations with label $k_0$ still has sample size $\tilde n = (1-o(1))n$ because we dropped $o(n)$ observations. We end up with a dataset with one fewer label, and almost the same sample size.
> Furthermore, if $y_i$ for $i=1,...,n$ are iid from a fixed distribution independent of $n,p$, if $P(y_{ik}=1)>0$ then by the weak law of large numbers, the probability to observe $\gamma*n$ responses $y_i$ with label $k$ converge to 1 for any $\gamma\in(0,P(y_{ik}=1))$. On the other hand, if $P(y_{ik}=1)=0$ then this label $k$ may be removed as it will not appear at all.
>
> For Assumption 2.4, we agree with you that imposing conditions on the MLE $\mathsf{\hat B}$ is not intrinsic. It is a reasonable assumption to us because (a) the event in Assumption 2.4 can be observed directly, and (b) because the goal of this paper is to study the property of $\mathsf{\hat B}$ provided it exists (it would not make sense to study such tests based on the multinomial MLE in settings where this MLE does not exist with high or positive provability; nothing can be done with MLE in the event that it does not exist).
> We suspect Assumption 2.4 could be reduced to "the dimension ratio p/n, covariance $\Sigma$ and xxx satisfy that xxx" as suggested by the referee, and as Candès and Sur (2021) did for binary logistic regression.
> However, such analysis for 3 or more classes does not exist yet (only partial answers are known, cf. lines 326-328 page 9)
> and a complete answer to this phase transition for multinomial logistic regression is beyond the scope of our paper.
>
> > I am not sure the convergence of some multinomial regression solver can be used as a rigorous theoretical indicator. The algorithm may converge (or be believed to converge) due to many reasons. Perhaps some better (numerical) criterion can be proposed by, e.g., checking the gradient and/or Hessian of the point of interest.
>
> We agree that checking the gradients/Hessian at the current iterate is a good criterion. We will clarify this paragraph: Our stance is to leverage (and not reinvent the wheel on) the well-studied convex solvers from optimization which often have their own built-in methods to assess convergence, for instance as returned in the ``success`` and ``message`` attributes of ``scipy.optimize.OptimizeResult`` for L-BFGS which is the default solver in scikit-learn. In practice, a definitive sign that convergence fails and the MLE does not exist is when the solver returns a $\hat B$ that perfectly separates the data in the sense that the corresponding softmax$(x_i^T\hat B)$ perfectly interpolates $y_i$ for all $i$. This holds for $K=1$: gradient descent is known to converge to such interpolator (max-margin classifier) on separable data (Soudry, Hoffer, Nacson, Gunasekar and Srebro 2018)., although we are not aware of similar works for $K>1$.
>
> > **Q2:** for the sake of presentation and use, it be helpful to present Theorem 2.2 and the estimation of $\Omega_{jj}$ in form of an algorithm.
>
> **A2:**
> Thanks.
> For the camera-ready version, we will include another equation in Theorem 2.2 with $\Omega_{jj}$ replaced by its estimator (2.5).
>
> > **Q3:** Almost nothing is mentioned for the proof of the theoretical results (Theorem 2.1 and 2.2): is the proof technically challenging or contains some ingredients and/or intermediate results that may be of independent interest?
>
> **A3:**
> The rigorous proof is non-trivial and, due to page limits, we decided to present it in the supplement. The first few pages of the supplement
> has a diagram explaining the relationship between the different theorems/lemmas, and how the lemmas in the supplement are combined to obtain the theorems stated in the main text.
>
> The main idea of the method used in the supplement consists of applying Lemma S5.3 to a carefully chosen function $F(\tilde z)$ taking values in the set of matrices with all singular values equal to 1, which is a new result of independent interest to exhibit asymptotic chi-square distributions in such problems.
> This poses unique challenges to ensure that the derivatives are well-behaved despite the normalization requiring all singular values to be equal to 1, and we solve some of these challenges specific to the cross-entropy loss in section S6 of the supplement. We expect some of these tools to be of independent interest and useful more broadly (beyond the cross-entropy loss) when one wishes to prove asymptotic chi-square distributions in related problems.

---

> > ### Comment · Reviewer_7JUy · 2023-08-14
> >
> > I thank the authors for the rebuttal and detailed clarifications.
> > And I maintain my positive rating on this paper.

---

### Official Review · Reviewer_FsXG · 2023-07-07

**Soundness:** 3 good
**Presentation:** 4 excellent
**Contribution:** 3 good
**Rating:** 6
**Confidence:** 3

**Summary:**

This paper studies the asymptotic distribution of the MLE of the multinomial logistic regression model when the sample size and the number of parameters are of the same order.  The validity of the asymptotic theories is evaluated through extensive simulation studies. The paper is overall very well written and the presentation is clear.

**Strengths:**

The paper is very well written, and the problem under consideration is of great interest. Theoretical results are sound and the numerical experiments are sufficient.

**Weaknesses:**

One of the motivating example of the paper is to study classification with 3 or more classes. And one of the most important goal in classification is the classification error or AUC. In my opinion, the impact of the proposed method on classification errors should be evaluated through simulation studies and some real data examples.

**Questions:**

1. Related to my question in the weakness section, one could use the proposed tests to exclude some noise covariates in the multinomial regression and see how much improvements can be achieved in terms of classification errors. Preferably, one should compare its performance to some existing methods.

2. I would strongly suggests applying the proposed methodology to some real-world benchmark data set. The statistical tests may provide additional insights and the classification errors can be evaluated through cross-validation.

3. The paper's theoretical framework appears to heavily rely on the seminal work of Sur and Candes (2019), because of which I still have some reservation on the novelty of the theory. To address concerns about the novelty of the theory, it would be wise to explicitly highlight the unique theoretical challenges posed by the multinomial logistic regression model compared to the more commonly studied binary logistic regression model.

---

> ### Author Rebuttal · Authors · 2023-08-06
>
> Thank you for the insightful and positive feedback.
>
> > **Q1**: Related to my question in the weakness section, one could use the proposed tests to exclude some noise covariates in the multinomial regression and see how much improvements can be achieved in terms of classification errors. Preferably, one should compare its performance to some existing methods.
>
> **A1:**
> The tests we propose provide strong guarantees for controlling Type I error: rejecting a null covariate (i.e., wrongly making a discovery regarding an insignificant covariate) happens with probability approximately $\alpha$ for any fixed $\alpha\in(0,1)$. However, this does not guarantee that the features that are not rejected should be safely removed from the model (which would require Type II error control). There could be instances where $B^*$ has many small coefficients on all features, so small that the test would not reject $H_0$ for all features; but overall the MLE $\hat B$ may still be more accurate (say, in prediction error) than random guess in such cases. In such situations, removing the features for which $H_0$ was not rejected would lead to a model with dimension 0 (with prediction probabilities $1/K$ for all classes) and would perform worse than the MLE for estimation of $B^*$ or prediction on a test set. In short, while the method suggested by the referee could be implemented, there are situations where using the MLE on the full model performs better.
>
> To our knowledge, there is no existing testing method with the same goal as ours for classification problems with 3 or more classes.
>
> > **Q2**: I would strongly suggests applying the proposed methodology to some real-world benchmark data set. The statistical tests may provide additional insights and the classification errors can be evaluated through cross-validation.
>
> **A2:**
> Thanks for the suggestion. We conducted a real data analysis by applying the proposed test to heart disease data from the UCI Machine Learning Repository. After standard data-cleaning processes, the dataset has 297 instances with 13 features, including age, sex, and other attributes. The response variable was transformed into 3 classes (0, 1, and 2) after converting 3 and 4 to 2. To demonstrate the validity of the proposed significance test, we generated a noise variable from a standard normal distribution, resulting in a dataset with 297 instances and 14 variables. We applied our test to this noise variable using both the proposed test and the classical test, repeating the experiment 10,000 times. The type I error of our proposed test is 0.0508, aligning well with the desired type I error of 0.05. In contrast, the classical test exhibits a type I error of 0.0734, significantly exceeding the desired rate of 0.05. These results confirm that the classical test tends to include noise variables, leading to false discoveries. On the other hand, our proposed test is more reliable and achieves the desired type I error.
>
> Regarding
> > The statistical tests may provide additional insights and the classification errors can be evaluated through cross-validation
>
> we refer to our response to the previous question, where we explained that the proposed method cannot be readily used to safely drop features from the model. Our method provides a strong guarantee for Type I error control, but this does not readily lead to improved classification error on unseen data.
>
> > **Q3**: The paper's theoretical framework appears to heavily rely on the seminal work of Sur and Candes (2019), because of which I still have some reservation on the novelty of the theory. To address concerns about the novelty of the theory, it would be wise to explicitly highlight the unique theoretical challenges posed by the multinomial logistic regression model compared to the more commonly studied binary logistic regression model.
>
> **A3:**
> The proposed theory is novel because it is different from the seminal work of [SC19] in at least three aspects:
>
> (i) The results in [SC19] only apply to binary classification problems, while our proposed theory also applies to classification problems with three or more classes.
>
> (ii) The techniques used in the proof are significantly different from those of [SC19]. Case-in-point: even for $K=2$, our argument does not rely on or involve the nonlinear system with 3 equations and 3 unknowns which is central to [SC19] for binary classification (or its generalization to $K\ge 3$ classes). The Approximate Message Passing proofs used by [SC19] are also not employed here. Our result for asymptotic multivariate normal and chi-square distributions is based on applying Lemma S5.3 of the supplement to a carefully chosen function $F(\tilde z)$ valued in the set of matrices with all singular values equal to 1, which is a new result of independent interest to exhibit asymptotic chi-square distributions in such problems. This further poses unique challenges to ensure that the derivatives are well-behaved despite the normalization requiring all singular values to be equal to 1, and we solve some of these challenges specific to the cross-entropy in section S6 of the supplement.
>
> (iii) From the computational side, all the quantities involved in our results can be readily computed from the data, while the relevant quantities in [SC19] require the estimation of the solutions to the nonlinear system mentioned in (ii). For $K=2$, this is not an issue, as [SC19] proposed the ProbeFrontier method to estimate the solutions to the nonlinear system.
> Any extension of their work to 3 or more classes (which, as far as we are aware, does not currently exist for the multinomial logistic model) would also require solving solutions for a new nonlinear system. We expect the ProbeFrontier method to break down since the new nonlinear system would now include several scalar parameters due to the matrix nature of $B^*$ (as opposed to the single unknown scalar $\gamma=Var[x_i^T\beta^*]$ in [SC19]).
>
> [SC19]: Sur and Candès (2019), PNAS

---

> > ### Comment · Reviewer_FsXG · 2023-08-21
> >
> > Thank you for your response. I have no further comments and will keep my initial score.

---

### Decision · Program_Chairs · 2023-09-21

**Decision:**

Accept (poster)

**Comment:**

This paper studies the asymptotic distribution of the maximum likelihood estimator for the multinomial logistic regression under the proportional growth regime (where the sample size and the number of unknowns are of the same order). The results developed in this paper generalized the previous theory by Sur et al. to accommodate classification with more than two classes. Asymptotic normality and asymptotic chi-square distributions have been established, which in turn suggest a new hypothesis test to test the significance of any given feature. The paper is well-written, and all reviewers recommend acceptance after the discussion phase. While the paper is mainly of theoretical interest, it would have broader impacts if the authors could demonstrate the efficacy of the proposed solution on more practical data.